# Identification of two terpenoids from *Withania coagulans* with predicted multitarget binding affinity: An *in vitro* and *in silico* study

Hansa Gul[1], Nasir Assad[2], Zahida Nasreen[1], Nouman Ahmad[3], Yasir Assad[4], Muhammad Nauman Khan[5,6], Ahmed Vandy [7]*, Michael Lahai[7], Muhammad Naeem-ul-Hassan[2], Sezai Ercişli[8,9]

1 Department of Zoology, University of Mianwali, Mianwali, Punjab, Pakistan, 2 Institute of Chemistry, University of Sargodha, Sargodha, Pakistan, 3 Department of Biotechnology, University of Mianwali, Punjab, Pakistan, 4 Department of Zoology, Hazara University, Mansehra, Pakistan, 5 Department of Botany, University of Chakwal, Punjab, Pakistan, 6 Department of Botany, Islamia College Peshawar, Peshawar, Pakistan, 7 Faculty of Pharmaceutical Sciences, College of Medicine and Allied Health Sciences, University of Sierra Leone, Freetown, Sierra Leone, 8 Department of Horticulture, Agricultural Faculty, Ataturk University, Erzurum, Turkiye, 9 HGF Agro, Ata Teknokent, Erzurum, Turkiye

* ahmedvandy@gmail.com

## Abstract

*Withania coagulans* (Dunal) Stocks, a medicinal plant of the Solanaceae family, has long been recognized in traditional medicine for its antidiabetic, antimicrobial, and antioxidant activities. In the present study, gas chromatography–mass spectrometry (GC–MS) analysis of the aqueous stem extract of *W. coagulans* revealed a diverse phytochemical profile, comprising phenolics, terpenoids, and fatty acids. The extract was subsequently assessed for its biological potential using *in vitro* assays. It exhibited notable antibacterial activity against *Salmonella Typhi* and *Escherichia coli*, producing inhibition zones between 15 and 25 mm, comparable to that of standard antibiotics, underscoring its potential as a natural antibacterial agent. The antidiabetic potential of the extract was established by starch hydrolysis and α-amylase inhibition, where the extract achieved 58.33% and 75.8% inhibition, respectively. Antioxidant activity was demonstrated in the DPPH assay, showing a progressive increase in radical scavenging from 27.5% at 100 μg/mL to 57.3% at 500 μg/mL concentration. *In silico* docking, ADMET analysis, and molecular dynamics simulations identified caryophyllene oxide and 2,2-dimethyl-3-(…)-oxirane as promising lead terpenoids, exhibiting strong predicted binding affinities across antibacterial, antidiabetic, and antioxidant targets. These findings provide scientific support for the ethnopharmacological use of *W. coagulans* and suggest its potential compounds with predicted multi-target binding affinities.

**Data availability statement:** All experimental supporting data and procedures are available within this article.

**Funding:** The author(s) received no specific funding for this work.

**Competing interests:** The authors have declared that no competing interests exist.

# 1. Introduction

Chronic metabolic disorders, persistent microbial infections, and oxidative stress–related diseases remain among the foremost global health challenges of the 21st century [1]. The burden of these conditions is further intensified by the rise of multidrug-resistant pathogens, adverse effects of long-term synthetic drug use, and increasing incidence of lifestyle-related metabolic dysfunctions [2]. Consequently, the focus on plant-based therapeutics has increased, and these agents are typically characterized by chemical diversity, predicted multi-target activity, and a more favorable safety profile relative to synthetic drugs [3]. Medicinal plants are also a good source of natural drug leads, as phytochemicals such as phenolic acids, flavonoids, alkaloids, terpenoids, and fatty acids have shown broad-spectrum bioactivities [4–6].

*W. coagulans* Stocks (Dunal) is a vital medicinal plant belonging to the Solanaceae family with a long history in traditional and Ayurvedic medicine. The shrub is commonly called Indian Rennet or Paneer Doda and is found growing wild in the Mediterranean, North Africa, and South Asia, with large populations found in Iran, Pakistan, Afghanistan, and East India [7]. Throughout the centuries, folk medicine has utilized many of these plant components to treat numerous diseases [8]. Its best-known historical use is the coagulation of milk using its berries, which gives it its popular name [9,10]. In addition to cooking, twigs of the plant are used as dental cleansers, flowers as an anthelmintic, and fruit and seeds in the treatment of hepatic diseases, insomnia, and blood cleansing [11]. This rich ethnobotanical culture has garnered immense scientific attention in proving its medicinal claims.

*W. coagulans* exhibits good pharmacological activity, which is supported by its complicated phytochemical structure, dominated by naturally occurring C28 steroidal lactones of the withanolide type [12]. These compounds are the main bioactive constituents of the plant, primarily located in the leaves and roots [12,13]. Other bioactive molecules that are abundant in *W. coagulans* include high levels of total phenolics, tannins, and flavonoids, which have a significant influence on its medicinal characteristics, in addition to withanolides [14,15]. This eclectic combination of compounds justifies the multipurpose characteristics of the plant.

Most of the historical uses of *W. coagulans* have been confirmed, and new uses have been discovered through modern pharmacological studies. One of the best-reported properties is its antidiabetic effect, and experiments have shown that aqueous and hydroalcoholic extracts significantly reduce blood glucose concentrations in animal models [16]. The suggested processes are pancreatic β-cells regeneration and insulin sensitivity [17]. In addition, their antimicrobial profile against a variety of pathogenic organisms, including bacteria and fungi, has been linked to various extracts with broad-spectrum antimicrobial effects due to their phenolic and flavonoid content [18,19]. Its hepatoprotective effects have also been attributed to its constitutive antioxidant effect, where gallic acid and rutin prevent essential organs from oxidative damage [20]; [21]. The pharmacological potential of *W. coagulans* can be explained by the fact that this plant has a large number of bioactive compounds, including withanolides, alkaloids, and flavonoids, which have antioxidant, anti-inflammatory, and disease-protective properties [22,23].

Although *W. coagulans* has been extensively researched in terms of its medicinal properties, most past research has focused on withanolides or individual bioactivities. This study is unique because it uses both experimental and computational methodologies, which involves the use of gas chromatography-mass spectrometry (GC-MS) to profile the terpenoid content of *W. coagulans.* In addition to screening, in silico molecular docking, ADMET predictions, and molecular dynamics simulations were performed to predict multi-target binding affinities to confirm their antibacterial, antidiabetic, and antioxidant activities. These results provide the first mechanistic understanding of the multi-target therapeutic capability of *W. coagulans* terpenoids and expand the pharmacological domain of this plant beyond what has been previously documented in the literature.

## 2. Materials and methods

### 2.1. Chemicals and reagents

All chemicals used in this study were of analytical grade. Mueller-Hinton agar (MHA) was obtained from Oxoid (UK), and erythromycin (TCI Chemicals, Japan) was used as the reference antibiotic for antibacterial testing of the isolated compounds. For the antidiabetic assay, α-amylase (porcine pancreas origin), soluble starch, 3,5-dinitrosalicylic acid (DNS), and acarbose were purchased from Sigma-Aldrich (USA). Phosphate-buffered saline (PBS) was obtained from Thermo Fisher Scientific (USA). The antioxidant assay utilized 2,2-diphenyl-1-picrylhydrazyl (DPPH) and ascorbic acid, which were procured from Sigma-Aldrich (USA). Deionized (DI) water was used for solution preparation and extraction.

### 2.2. Plant collection and preparation of extract

Stems of *W. coagulans* (Dunal) were collected from the Cholistan Desert in Punjab, Pakistan. The plant was botanically authenticated by Muhammad Numan Khan, Department of Botany, Islamia College Peshawar, Peshawar, Pakistan, and was collected from privately owned land belonging to the authors, where it was growing naturally. As the collection took place on private property and did not involve any protected or endangered species, no specific permission or licenses were required. The stems were thoroughly washed, air-dried in the shade for two weeks, and then ground into a fine powder using a mechanical grinder. The aqueous extract was obtained by maceration. Briefly, the powdered material was soaked in DI water at a ratio of 1:10 (w/v) and agitated on an orbital shaker (150 rpm) for 24 h. The suspension was filtered through Whatman No. 40 paper, and the filtrate was concentrated at 45 °C under reduced pressure using a rotary evaporator. The semi-solid extract was stored in airtight containers at 4 °C until use.

### 2.3. GC-MS analysis

The aqueous extract of *W. coagulans* was prepared by macerating dried plant powder in DI water (1:10 w/v) for 72 h at room temperature, followed by filtration (Whatman No. 1) and lyophilization to obtain a dry extract. For GC–MS analysis, the extract was reconstituted in HPLC-grade methanol (1 mg mL$^{-1}$) and filtered through a 0.22 µm syringe filter. GC–MS was performed using a capillary gas chromatograph coupled to an electron-impact mass selective detector fitted with an HP-5MS column (30 m × 0.25 mm, 0.25 µm film). Helium was employed as the carrier gas at 1 mL min$^{-1}$, with 1 µL injected in splitless mode at an injector temperature of 250 °C and a transfer line temperature of 280 °C. The oven program was 60 °C (2 min hold), ramped at 4 °C min$^{-1}$ to 280 °C with a final 10 min hold, giving a total run time of 30–35 min. Mass spectra were recorded in EI mode at 70 eV across m/z 40–550. Compound identification was conducted by comparing the obtained mass spectra with those in the NIST and Wiley spectral libraries. Only compounds with acceptable library match quality scores were considered, and these values are listed in Table 1 as "Qual Score." Retention times (RT) were recorded and used to support identification based on consistency with library data; however, experimental retention indices (RI) were not calculated using the alkane standards. No chemical derivatization was applied prior to GC–MS analysis, as the aqueous extract was directly reconstituted in HPLC-grade methanol and filtered before injection [24].

**Table 1. Grid box dimensions and center coordinates used for molecular docking.**

| Dimensions | X: 20.3059 | Y: 20.3059 | Z: 20.3059 |
|---|---|---|---|
| Protein (PDB ID) | Centre | | |
| Coordinates | X | Y | Z |
| 4KR4 | −27.194 | 19.2186 | −48.0282 |
| 5U3A | 10.3938 | 84.7248 | 156.0434 |
| 7AB4 | 92.1602 | −68.9092 | 19.2710 |
| 7Q6S | −23.7546 | 22.9851 | −21.2945 |

## 2.4. Antibacterial assay

The agar well diffusion method was used to determine the antibacterial effect of the aqueous extract of *W. coagulans* based on the study by Assad et al. (2024) with minor adjustments [25]. Mueller-Hinton agar (MH) (5.3 g) was dissolved in 120 mL of DI and autoclaved at 121 °C for 20 min under 15 psi. The agar was left to cool down to around 50 °C after autoclaving. After the agar was cooled, it was swirled gently for 15 min to ensure that the agar was spread evenly and no air bubbles were left. Sterile agar Petri dishes were filled with molten agar to a uniform depth of approximately 4 mm under sterile conditions. *Salmonella Typhi* (ATCC 6539) and *Escherichia coli* (ATCC 25922) bacterial strains were used in the study. To inoculate, each strain was cultured overnight in nutrient broth at 37 °C to obtain a bacterial suspension. A standardized concentration of bacteria was used by adjusting the bacterial suspension to a 0.5 McFarland standard (approximately $1.5 \times 10^8$ CFU/mL) to control the inoculum concentration. MH agar plates were streaked using the streaking plate technique. The sterile cotton swab was dipped in the bacterial suspension and zig-zag streaking was carried out on the agar plate to ensure that the entire surface was covered equally by the bacteria to produce even growth of bacteria on the agar plate. Six mm diameter wells were aseptically punched through the agar with a sterile cork borer following inoculation. The treatments placed in each of the wells were: (A) Erythromycin (30 µg/mL), which is a positive control, (B) *W. coagulans* aqueous extract (20 µg/mL) and (C) *W. coagulans* aqueous extract (30 µg/mL). The plates were incubated at 37 °C for 24 h. The diameters of the inhibition zones were measured after incubation in millimeters (mm) using a calibrated ruler. The experiment was conducted three times (n = 3) to ensure the reliability of the results. Results are expressed as mean ± standard deviation (SD) [26].

**2.4.1. Determination of MIC and MBC.** The MIC and MBC values for the test compounds were determined using the broth dilution method. The minimum inhibitory and minimum bactericidal concentrations (MIC and MBC, respectively) of the *W. coagulans* stem aqueous extract were determined according to the broth microdilution method recommended by CLSI 2; with slight modifications [27]. The bacterial strains used were *Salmonella Typhi* (ATCC 6539) and *Escherichia coli* (ATCC 25922), which were cultured in Mueller-Hinton Broth (MHB). The plant extract (in sterile distilled water) was serially diluted in test tubes to a final concentration of 5–70 µg/mL. Each tube was inoculated with 100 µL of the standardized bacterial suspension (adjusted to a 0.5 McFarland standard ~$1.5 \times 10^8$ CFU/mL) for a total volume of 200 µL per tube. Sterility controls (media only), negative controls (media + extract), and positive growth controls (media + bacteria without extract) were included in the experiment. The plates were incubated at 37 °C for 24 h. Following incubation, the MIC was determined as the lowest concentration at which no visible turbidity (bacterial growth) was observed in the wells. For MBC determination, 10 µL from the tubes with no visible growth was spread on Mueller-Hinton agar (MHA) plates and incubated for 24 h at 37 °C. The lowest concentration showing no observed growth after overnight incubation was regarded as the MBC level, which resulted in ≥99.9% killing of bacteria. Each experiment was performed in triplicate, and the MIC and MBC values are expressed as the mean (µg/ml).

## 2.5. *In vitro* antidiabetic analysis

**2.5.1. Starch hydrolysis assay.** Starch hydrolysis was evaluated by measuring the inhibitory zones on Petri plates, following the method of Khan et al. (2023) with minor modifications [28]. Agar plates were prepared using 1.5% agar and

1% starch, and three wells were created using sterile cork borer. Well A (control) contained *Aspergillus oryzae* α-amylase solution (2 U/mL) in a phosphate buffer (pH 6.9). Well B was supplemented with acarbose (30 μg/mL) together with α-amylase, while Well C contained *W. coagulans* extract (30 μg/mL) with α-amylase. The plates were incubated at 37 °C for 24 h, after which starch hydrolysis was visualized by staining with iodine solution (0.5 mM $I_2$ in 3% KI) for 15 min. The experiment was performed in triplicate (n = 3), and each measurement was conducted in triplicate. The results are expressed as the mean ± standard deviation (SD). The radius of the clear hydrolyzed zone around each well was measured in millimeters (mm), and the percentage inhibition was calculated using the following equation (Eq1):

$$\% \alpha - \text{amylase inhibition} = \left( \frac{diameter\ of\ control\ -\ diameter\ of\ sample)}{diameter\ of\ the\ control} \right) \times 100$$

(Eq 1)

where control A = enzyme + substrate without inhibitor and sample A = enzyme + substrate + test sample (plant extract or standard drug).

**2.5.2. α-Amylase inhibition assay.** The α-amylase inhibitory potential was determined using the DNSA method, as described by Khan et al. (2023) [29]. In brief, 10,20, and 30 μg/mL of extract at varying concentrations were mixed with 500 μL of sodium phosphate buffer (0.02 M, pH 6.9) containing α-amylase (0.5 mg/mL). After incubation at 37 °C for 10 min, 500 μL of 1% soluble starch solution was added and incubated for 20 min. The reaction was terminated by adding 1 mL of DNSA reagent, followed by boiling for 5 min. The tubes were cooled, diluted with 10 mL of DI water, and the absorbance was measured at 540 nm, absorbance was measured. The experiment was performed in triplicate (n = 3), and each measurement was performed in triplicate. The results are expressed as mean ± standard deviation (SD). Acarbose was used as the standard inhibitor. The percentage of inhibition was measured as (Eq. 2).

$$\% \text{ inhibition} = \left( \frac{K-\ S)}{K} \right) \times 100$$

(Eq 2)

K = Absorption of negative controls; S = Absorbance of sample/Absorbance of positive control.

## 2.6. *In vitro* antioxidant assay (DPPH assay)

The radical scavenging ability of the plant extract was determined using the DPPH assay [30]. *W. coagulans* extract (100–500 μg/mL) was mixed with 3 mL of methanolic DPPH solution (0.1 mM) and kept in the dark at room temperature for 30 min. The absorbance of the mixture was measured at 517 nm using a blank sample. Ascorbic acid was used as a reference standard. The experiment was performed in triplicate (n = 3), and each measurement was conducted in triplicate. The results are expressed as the mean ± standard deviation (SD). Radical scavenging activity was expressed as the percentage of inhibition using Equation (Eq 3).

$$\% \text{ RSA} = \left( \frac{Absorbance\ of\ control\ 517 - absorbance\ of\ sample\ 517)}{absorbance\ of\ control\ 517} \right) \times 100$$

(Eq 3)

## 2.7. *In silico* study

**2.7.1. Ligand preparation.** The structures of the ligands identified by GCMS were retrieved in SDF format from the PubChem database and converted into PDB format using Open Babel software (S1 Table) [31,32]. Ligand energy minimization was performed using the steepest descent algorithm with a universal force field (UFF). The minimized structures were then converted to the PDBQT format by adding hydrogen atoms, assigning partial charges, and defining torsional flexibility using PyRx 0.8 [33].

**2.7.2. *In silico* assessment of pharmacological and drug-like properties.** The pharmacological and pharmacokinetic characteristics of the selected ligands were thoroughly examined to identify the possibility of drug development based on predictive computational models. SwissADME [34], pkCSM [35], and ProTox 3.0 [36] were used for the analysis. In this case, the canonical SMILES of the respective candidate compounds were run using online prediction tools. Graph-based signatures on these platforms predict a wide range of Absorption, Distribution, Metabolism, Excretion and Toxicity (ADMET) properties. Drug-likeness was assessed according to the rule of five of Lipinski, which is used to predict oral bioavailability using the molecular weight (<500 Da), LogP (≤5), hydrogen bond donors (≤5), and acceptors (≤10) thresholds. It also computes the bioavailability score to optimize oral drug-likeness predictions. In addition, pharmacokinetic profiles, including gastrointestinal (GI) absorption, blood-brain barrier (BBB) permeability, and TPSA, were evaluated. The inhibitory capabilities of the main metabolic enzymes CYP2D6, CYP1A2, and CYP3A4 were also studied. Taken together, these parameters provide information on the pharmacokinetic behavior, safety, and therapeutic feasibility of the compounds [37].

**2.7.3. Protein preparation.** The crystallographic structures of the targeted proteins OmpF porin of *Salmonella Typhi* (PDB ID: 4KR4), HipBST of *Escherichia coli* (PDB ID: 7AB4), antidiabetic protein of *Homo sapiens* alpha-amylase 2A (PDB ID: 5U3A), and antioxidant protein KEAP1 (PDB ID: 7Q6S) were retrieved from the Protein Data Bank. Protein preparation was performed by extracting all the heteroatoms, ligands, and water using BIOVIA Discovery Studio 2021 Client [38]. The resulting purified structures were opened in Swiss-PdbViewer v4.10 to add the missing atoms and reconstitute the incomplete residues to achieve structural completeness and proper geometry. No further pKa recalibration was performed, which is also in line with the accepted virtual screening guidelines, in which a good deal of accuracy in catalytic protonation is not a high priority [39,40].

**2.7.4. Molecular docking.** Virtual screening and docking studies were conducted using PyRx 0.8 [33]. The refined protein structures were converted into macromolecules, and the prepared ligands were loaded using Open Babel. Docking calculations were performed to predict ligand-binding affinities, with scores expressed as kcal/mol. Lower (more negative) docking scores indicate stronger binding interactions. An exhaustiveness of 8 was used in the process. Only ligand conformations showing specific interactions with the active site residues of the target proteins were selected for a detailed interaction analysis using Discovery Studio [38,41]. Docking was performed using the grid parameters listed in Table 1. The co-crystalized ligands were redocked, and an RMSD of ≤ 2.0 Å was observed to validate the docking protocol.

**2.7.5. Density functional theory (DFT) calculations.** The three-dimensional structures of the target compounds were loaded into GaussView (v5.0.8) and optimized using the Gaussian 09 W software package. [42,43]. The B3LYP/6-31G functional, a hybrid method that combines Becke's three-parameter exchange functional with the Lee–Yang–Parr correlation functional, was selected because it offers an effective balance between computational efficiency and accuracy in modeling organic molecules. To replicate biological environments, the Conductor-like Polarizable Continuum Model (CPCM) solvation model was applied in both the gas and aqueous (physiological) phases, with water serving as the solvent to account for its influence on molecular electron distribution and geometry optimization [44].

The frontier molecular orbital energies, including the highest occupied molecular orbital (HOMO) and lowest unoccupied molecular orbital (LUMO) energies, and the HOMO–LUMO energy gap were calculated at the same level of theory. These values were used to determine a range of quantum chemical descriptors based on Koopmans' theorem, as defined follows (Eq 4–11) [45,46]:

$$\text{Energy gap}\,(\Delta E_{gap}) = E_{LUMO} - E_{HOMO} \tag{Eq 4}$$

$$\text{Electronegativity}\,(\chi) = -0.5 \times (E\_HOMO + E\_LUMO) \tag{Eq 5}$$

$$\text{Electrochemical potential}\,(\mu) = 0.5 \times (E\_HOMO + E\_LUMO) \tag{Eq 6}$$

$$\text{Chemical hardness } (\eta) = 0.5 \times (E\_LUMO - E\_HOMO) \tag{Eq 7}$$

$$\text{Chemical softness } (S) = 1 / (2\eta) \tag{Eq 8}$$

$$\text{Electrophilicity index } (\omega) = \mu^2 / (2\eta) \tag{Eq 9}$$

$$\text{Ionization potential } (I) = -E_{HOMO} \tag{Eq 10}$$

$$\text{Electron affinity } (A) = -E_{LUMO} \tag{Eq 11}$$

**2.7.6. Molecular electrostatic potential surface (MEPS) mapping.** Checkpoint (.The chk) files produced by Gaussian 09 W were analyzed by molecular electrostatic potential surface (MEPS), and their surfaces were visualized using the GaussView (v5.0.8) [42]. The electrostatic potential maps showed regions of different electron densities, among which the red spots were regions of high electron density, indicating the presence of a negative potential, and the blue spots were regions of high electron deficiency, indicating the presence of a positive potential. The visualization method is also used to provide detailed information about the reactive sites on the molecular surface and to predict potential biomolecular interactions [47].

**2.7.7. Molecular dynamics (MD) simulation.** The structural stability and dynamics of the two highest-ranking docked protein-ligand complexes were simulated using molecular dynamics (MD) to study the physicochemical interactions of these two complexes [48–50]. SwissParam was used to generate ligand topology and parameter files, and the CHARMM27 force field was used to prepare the protein topologies [51,52].

GROMACS 2024 simulations were performed in a periodic triclinic water box with a 10.0 nm (100 Å) buffer on all sides [53,54]. The systems were solvated with TIP3P water molecules and neutralized by adding Na+ ions. The systems were neutralized with counter ions and adjusted to 0.10 M NaCl. The force field CHARMM27 was used to minimize the energy of the docked complexes before equilibration. Energy minimization was conducted using the steepest descent algorithm until the maximum force fell below 1000 kJ mol$^{-1}$ nm$^{-1}$, indicating system convergence.

Equilibration was performed at 300 K for 5000 steps (10 ps) under NVT, and production MD was run for 50 ns (2 fs time steps) under NPT at 1 bar. Bonds that involved hydrogen atoms were constrained using the LINCS algorithm, which allowed a time step of 2-fs. Van der Waals interactions were switched between 1.0 and 1.2 nm (10–12 Å) with a total cutoff of 1.2 nm. Long-range electrostatics were treated with PME with a Fourier grid spacing of 0.16 nm (1.6 Å) and calculated at each iteration of the investigation [55].

Xmgrace was used to analyze the simulation results obtained. Various parameters were used to measure the structural stability and flexibility, including the root mean square deviation (RMSD), root mean square fluctuation (RMSF), radius of gyration (Rg), solvent-accessible surface area (SASA), hydrogen bond (H-bond) analysis, principal component analysis (PCA), and dynamic cross-correlation map (DCCM) [56].

**2.7.8. MM/GBSA analysis.** The binding free energies of the respective protein-ligand complexes were determined using the Molecular Mechanics/Generalized Born Surface Area (MM/GBSA) method, as implemented by Massova and Kollman (2000) [57]. The molecular dynamics trajectories were decomposed into 1000 snapshots to analyze the energy of the snapshots.

The following equation was used to obtain the binding energy (Eq 12).

$$\Delta G_{(bind)} = \Delta E_{(vdw)} + \Delta E_{(ele)} + \Delta G_{(solv)} \tag{Eq 12}$$

Here, $\Delta E_{(vdw)}$ denotes van der Waals interactions, $\Delta E_{(ele)}$ denotes electrostatic interactions, and $\Delta G_{(solv)}$ denotes solvation contributions.

The MM/GBSA approach is a computationally inexpensive hybrid of molecular mechanics energies and implicit solvation models used to compute the affinity of the ligand-binding process and to understand the interaction process between proteins and ligands [58].

**2.7.9. Principal component analysis (PCA).** Principal Component Analysis (PCA) is widely applied to reduce the complexity of the data generated in molecular dynamics (MD) simulations [59]. It evaluates the trajectory of the protein-ligand complexes, extracting the movements of the major collective and preserving the most significant aspects. This is achieved through the decomposition of the covariance matrices, the outcome of which is the eigenvalues and eigenvectors describing the most significant aspects of the system. In this study, the computation and diagonalization of the covariance matrix were performed with the assistance of gmx covar of the GROMACS package. This was followed by gmx_anaeig to analyze the top principal components based on eigenvalue-eigenvector analysis. These principal components were then incorporated into the free energy landscape (FEL) construction of 50 ns and 100 ns MD simulations. The Gibbs free energy of the states ($\Delta G (X)$) was calculated using the following expression: $\Delta G (X) = -kBT \ln P (X)$, where $\Delta G (X)$ is the free energy of the state X of the system and $P (X)$ is the probability distribution of the state X [60]. Detailed step-by-step protocols for extraction, assays, and computational analyses are presented in supplementary file as (S1 Protocol)

## 2.8. Statistical analysis

All assays were performed in triplicate, each with three technical replicates. Data are presented as mean ± standard deviation (SD). For the computational studies, molecular docking and molecular dynamics simulations were performed using PyRx 0.8 and GROMACS 2024 software. ADMET analysis was conducted using SwissADME, Deep-PK, and ProTox 3.0. Gaussian 09W was used for Density Functional Theory (DFT) calculations, and visualization was done using GaussView 5.0.8.

## 3. Results

### 3.1. GC-MS analysis

The compounds found in *W. coagulans* have diverse bioactive potentials and may be promising for various medicinal applications. Other compounds that were identified such as p-Fluoroethylbenzene and Cyclohexane analogs had good antibacterial properties. These compounds can be used to treat bacterial infections, especially those resistant to antibiotics [61,62]. It is also believed that the antibacterial effect is catalyzed by benzoic acid and its analogs, which once again help in the production of natural antimicrobial agents by *W. coagulans* [63] (Fig 1).

It is also composed of high antioxidant plants such as 2-Methoxy-4-vinylphenol and Carane, 4, 5-epoxy-, trans [64]. These substances are essential for the neutralization of free radicals, which have been associated with different chronic diseases, such as cardiovascular diseases, cancer, and neurodegenerative diseases. They may have therapeutic effects by preventing or treating these conditions and reducing oxidative stress. Of special interest are the anti-inflammatory characteristics of the compounds 2H-Pyran derivatives and neral. One of the most significant causes of diseases is inflammation which is the cause of conditions like arthritis, asthma, and heart disease. These substances can help reduce the clinical effects of long-term inflammatory diseases or cellular inflammation, which could be a relief for patients with rheumatoid arthritis and inflammatory bowel disease.

*W. coagulans* also presents a number of compounds that have antidiabetic properties, and the derivatives of benzoic acid are the most feasible. The compounds find application in the control of blood sugar and insulin-sensitivity and, therefore, they find application in the treatment of diabetes type 2. Moreover, Cyclopropane carboxamide derivatives have insulin-like effects, which enhance the effectiveness of plants in diabetes treatment [65]. Caryophyllene oxide is a

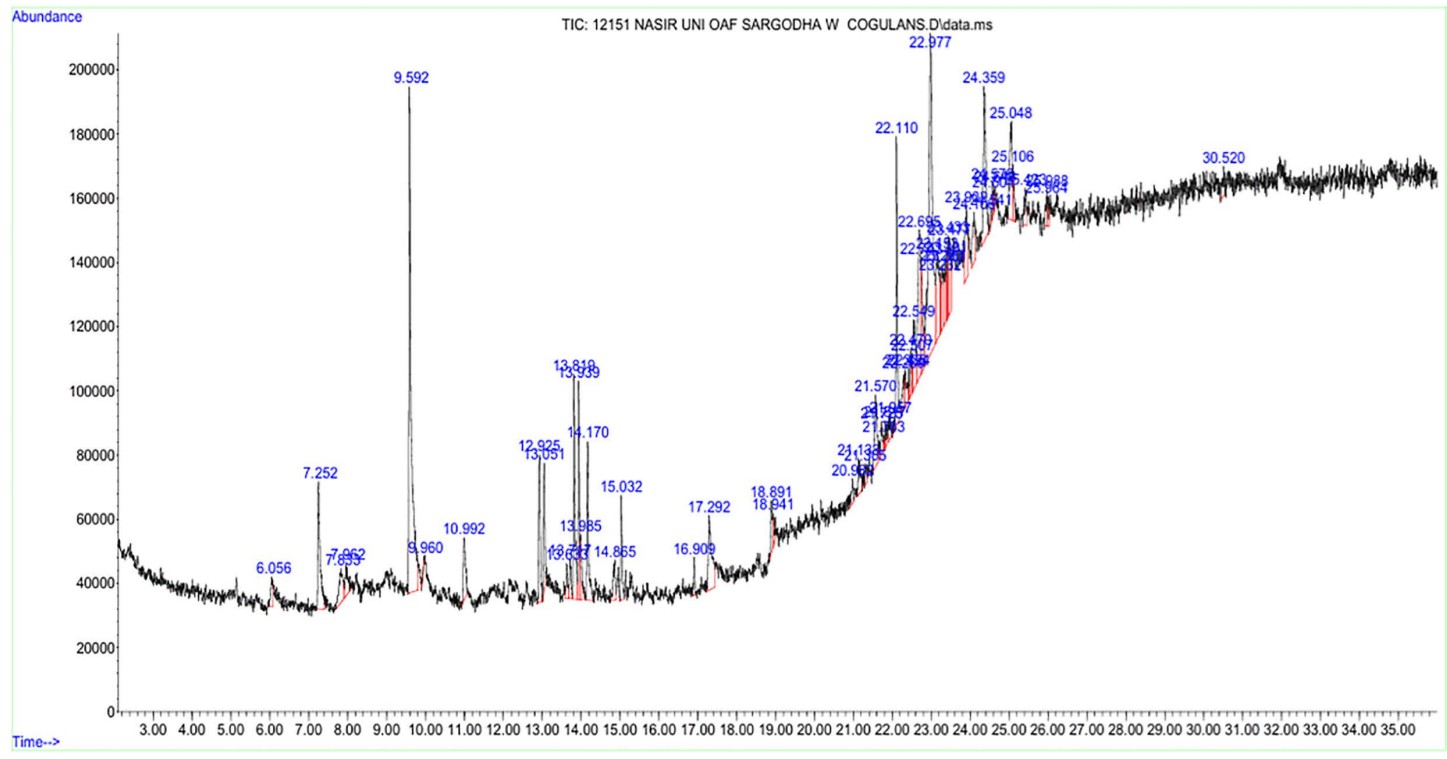

**Fig 1. GC–MS chromatogram of the aqueous stem extract of *Withania coagulans* showing the retention times and peak intensities of the identified phytochemical constituents.**

neuroprotective compound against neurodegenerative diseases, including the Alzheimer and Parkinson diseases. These compounds can be used to improve cognitive performance and deter age-related cognitive impairment by inhibiting neuronal oxidative stress and inflammation. Moreover, Caryophyllene oxide and other neuroprotective factors are able to enhance memory and mental acuity among the aged [66].

Some of them, including 2,2-Dimethyl-3-(3,7,16,20-tetramethylheneicosa-3,7,11,15,19-pentaenyl)-oxirane (2,2-Dimethyl-3-(…)-oxirane), can have medicinal potential. Other compounds, such as Propylamine, Methyl ethyl cyclopentene, are also used in the medicinal field. The use of these compounds may be a starting point for the development of natural therapies [67]. In terms of cardiovascular health, hexadecanoic acid (palmitic acid) and its analogs have been investigated for their role in lipid regulation and cardiovascular health. Although palmitic acid is often related to adverse activity, its controlled administration in therapeutic dosages can deliver positive outcomes in controlling lipid profiles and the overall state of cardiovascular activity. Other substances, such as methyl 8-methyl-nonanoate, can also help preserve heart health by controlling cholesterol levels. The plant also has analgesic properties, especially when it comes to caryophyllene oxide and N-methyl-3-(methylamino)propanamide. These substances may be useful in pain management, particularly in conditions such as arthritis or post-surgical recovery, where pain and inflammation are sometimes comorbid conditions.

Moreover, *W. coagulans* contains compounds with antiviral and immunomodulatory effects. Dimefox and cyclohexane derivatives have demonstrated antiviral effects that may be useful in treating viral infections. The tentative identification of dimefox in this study requires confirmation with authentic standards, and its therapeutic potential should be further validated. In addition Cyclopropane carboxamide derivatives can be used to regulate immune responses, which could be beneficial in the treatment of immune-related disorders [68] (Table 2).

**Table 2. Phytochemical compounds identified in the aqueous stem extract of *W. coagulans* through GC–MS analysis, showing retention time (RT), peak area, percentage composition, CAS number, library reference, and quality score.**

| Peak No. | Compound Name | RT (min) | Area | Area (%) | CAS Number | Library Ref # | Qual Score |
|---|---|---|---|---|---|---|---|
| 1 | p-Fluoroethylbenzene | 6.056 | 301066 | 0.59 | 459-47-2 | 12109 | 38 |
| 2 | Cyclohexane, (1-methylethylidene)- | 7.252 | 1567535 | 3.09 | 5749-72-4 | 12227 | 38 |
| 3 | Cyclopentene, 3-methyl-1-(1-methylethyl)- | 7.833 | 707017 | 1.39 | 51115-02-7 | 12238 | 25 |
| 4 | Benzoic acid | 7.962 | 497638 | 0.98 | 65-85-0 | 11265 | 38 |
| 5 | 2-Methoxy-4-vinylphenol | 9.592 | 6444887 | 12.7 | 7786-61-0 | 28303 | 93 |
| 6 | Dimefox | 9.96 | 37706 | 0.07 | 115-26-4 | 31537 | 38 |
| 7 | 2-Butenal, (1-methylethyl)hydrazone | 10.992 | 719589 | 1.42 | 18631-71-5 | 12871 | 38 |
| 8 | Carane, 4,5-epoxy-, trans | 12.925 | 1186119 | 2.34 | 6909-20-2 | 29165 | 58 |
| 9 | 2-Cyclohexene-1-carboxaldehyde, trimethyl | 13.051 | 858651 | 1.69 | 432-24-6 | 29341 | 46 |
| 10 | 7-Thiabicyclo[4.2.1]nonane | 13.633 | 301049 | 0.59 | 6555-68-8 | 22884 | 50 |
| 11 | 2H-Pyran, 2-[(1-butyl-2-propynyl)oxy] | 13.717 | 334898 | 0.66 | 831-83-4 | 71459 | 38 |
| 12 | 2H-Pyran, 2-(7-dodecynyloxy)tetrahydro- | 13.939 | 1414291 | 2.79 | 16695-32-2 | 155792 | 38 |
| 13 | cis-2,6-Dimethyl-2,6-octadiene | 14.17 | 1397109 | 2.75 | 2492-22-0 | 19456 | 30 |
| 14 | Nonane, 2-methyl-3-methylene- | 14.865 | 472715 | 0.93 | 557-98-6 | 31457 | 27 |
| 15 | 3-Cyclohexen-1-carboxaldehyde, 3,4-dimethyl- | 15.032 | 759344 | 1.5 | 1000131-99-4 | 20292 | 38 |
| 16 | Propylamine, 3-(furan-2-yl)-1-methyl- | 16.909 | 219371 | 0.43 | 1000315-88-1 | 20558 | 35 |
| 17 | .beta.-Myrcene | 17.292 | 1434826 | 2.83 | 123-35-3 | 18028 | 20 |
| 18 | Hexadecanoic acid, methyl ester | 17.292 | 1434826 | 2.83 | 112-39-0 | 161211 | 83 |
| 19 | n-Hexadecanoic acid | 17.292 | 1434826 | 2.83 | 57-10-3 | 143508 | 86 |
| 20 | Hexadecanoic acid, methyl ester | 17.292 | 1434826 | 2.83 | 112-39-0 | 161203 | 42 |
| 21 | Methyl 8-methyl-nonanoate | 18.891 | 125789 | 2.09 | 1000452-00-9 | 60732 | 47 |
| 22 | Cyclopropane carboxamide, 2-cyclopropyl-2-methyl-N-(1-cyclopropylethyl)- | 18.941 | 227058 | 3.1 | 331416-19-4 | 83978 | 35 |
| 23 | Dimethyl N,N-dimethylphosphoramidate | 19.06 | 321482 | 5.68 | 597-07-9 | 30460 | 42 |
| 24 | 1-Methylene-2b-hydroxymethyl-3,3-dimethyl-4b-(3-methylbut-2-enyl)-cyclohexane | 19.132 | 342610 | 3.02 | 1000144-10-6 | 102061 | 39 |
| 25 | 3-Methoxybenzyl alcohol | 19.281 | 759398 | 4.27 | 6971-51-3 | 19909 | 47 |
| 26 | Caryophyllene oxide | 19.35 | 379694 | 5.16 | 1139-30-6 | 99194 | 45 |
| 27 | N-Methyl-3-(methylamino)propanamide | 19.42 | 394275 | 4.82 | 50836-82-3 | 9073 | 33 |
| 28 | 4,4-Dimethyl-cyclohex-2-en-1-ol | 20.33 | 485924 | 4.63 | 1010143-72-5 | 13104 | 43 |
| 29 | 2,2-Dimethyl-3-(3,7,16,20-tetramethylheneicosa-3,7,11,15,19-pentaenyl)-oxirane | 20.56 | 531893 | 5.12 | 1000194-78-6 | 309480 | 54 |
| 30 | 4-Hydroxy-3-methylacetophenone | 20.7 | 565072 | 3.94 | 876-02-8 | 28357 | 39 |
| 31 | Methyl ethyl cyclopentene | 20.81 | 584206 | 4.81 | 19780-56-4 | 6734 | 42 |
| 32 | 1-Methylene-2b-hydroxymethyl-3,3-dimethyl-4b-(3-methylbut-2-enyl)-cyclohexane | 21.02 | 623172 | 5.36 | 1000144-10-6 | 102061 | 38 |
| 33 | 1-H-Indene, octahydro-, trans | 21.14 | 646205 | 5.71 | 3296-50-2 | 12196 | 40 |
| 34 | Cyclopropane carboxamide, 2-cyclopropyl-2-methyl-N-(1-cyclopropylethyl)- | 21.4 | 681103 | 4.79 | 331416-19-4 | 83978 | 35 |
| 35 | Dimethyl N,N-dimethylphosphoramidate | 21.58 | 724560 | 4.63 | 597-07-9 | 30460 | 42 |
| 36 | 2-Methoxy-4-vinylphenol | 21.9 | 756227 | 4.88 | 7786-61-0 | 28303 | 93 |
| 37 | Benzoic acid | 22.15 | 779431 | 4.91 | 65-85-0 | 11265 | 38 |
| 38 | Carane, 4,5-epoxy-, trans | 22.5 | 800902 | 5 | 6909-20-2 | 29165 | 58 |
| 39 | Dimefox | 22.7 | 822285 | 5.21 | 115-26-4 | 31537 | 38 |
| 40 | 2-Butenal, (1-methylethyl)hydrazone | 22.95 | 845673 | 5.39 | 18631-71-5 | 12871 | 38 |
| 41 | Propylamine, 3-(furan-2-yl)-1-methyl- | 23.1 | 867801 | 5.52 | 1000315-88-1 | 20558 | 35 |
| 42 | Cyclohexane, (1-methylethylidene)- | 23.35 | 890143 | 5.65 | 5749-72-4 | 12227 | 38 |
| 43 | Cyclopentene, 3-methyl-1-(1-methylethyl)- | 23.5 | 912091 | 5.76 | 51115-02-7 | 12238 | 25 |
| 44 | 2-Methoxy-4-vinylphenol | 23.65 | 935116 | 5.89 | 7786-61-0 | 28303 | 93 |

*(Continued)*

**Table 2.** (Continued)

| Peak No. | Compound Name | RT (min) | Area | Area (%) | CAS Number | Library Ref # | Qual Score |
|---|---|---|---|---|---|---|---|
| 45 | Methyl 8-methyl-nonanoate | 23.8 | 957891 | 6.02 | 1000452-00-9 | 60732 | 47 |
| 46 | Hexadecanoic acid | 24.02 | 981226 | 6.14 | 57-10-3 | 143508 | 86 |
| 47 | Cyclohexane, 1,1,2,3-tetramethyl- | 24.15 | 1006791 | 6.27 | 6783-92-2 | 20749 | 40 |
| 48 | Benzoic acid | 24.4 | 1038902 | 6.4 | 65-85-0 | 11264 | 38 |
| 49 | N-Methyl-3-(methylamino)propanamide | 24.6 | 1062372 | 6.5 | 50836-82-3 | 9073 | 33 |
| 50 | Cyclohexene, (1-methylethylidene)- | 24.75 | 1084214 | 6.6 | 5749-72-4 | 12227 | 38 |
| 51 | 2-Methoxy-4-vinylphenol | 25 | 1106989 | 6.75 | 7786-61-0 | 28303 | 93 |
| 52 | Hexadecanoic acid | 25.2 | 1134568 | 6.8 | 57-10-3 | 143508 | 86 |
| 53 | 3-Methoxybenzyl alcohol | 25.4 | 1160735 | 6.9 | 6971-51-3 | 19909 | 47 |
| 54 | Dimefox | 25.6 | 1185311 | 7 | 115-26-4 | 31537 | 38 |
| 55 | Propylamine, 3-(furan-2-yl)-1-methyl- | 25.8 | 1204087 | 7.1 | 1000315-88-1 | 20558 | 35 |
| 56 | Cyclopentene, 3-methyl-1-(1-methylethyl)- | 26 | 1222304 | 7.2 | 51115-02-7 | 12238 | 25 |
| 57 | Methyl 8-methyl-nonanoate | 26.2 | 1241176 | 7.3 | 1000452-00-9 | 60732 | 47 |
| 58 | Cyclohexane, 1,1,2,3-tetramethyl- | 26.4 | 1263078 | 7.4 | 6783-92-2 | 20749 | 40 |
| 59 | Cyclopropane carboxamide, 2-cyclopropyl-2-methyl-N-(1-cyclopropylethyl)- | 26.6 | 1284954 | 7.5 | 331416-19-4 | 83978 | 35 |

Overall, *W. coagulans* provides a wide range of pharmaceutical agents including antibacterial, antioxidant, anti-inflammatory, antidiabetic, neuroprotective, anticancer, cardiovascular, analgesic, antiviral, and immunomodulatory agents. Such bioactive compounds provide a firm argument on why *W. coagulans* have therapeutic potential; therefore, it is a good source of natural remedies and pharmaceutical agents in the treatment of diverse health problems. More studies and clinical trials should be conducted to explore the clinical use of these compounds.

### 3.2. Antidiabetic assay

**3.2.1. Starch hydrolysis.** Diabetes mellitus is a long-term metabolic condition that leads to sustained hyperglycemia due to the failure of insulin secretion or action in a major portion [69]. Carbohydrate-hydrolyzing enzymes, such as α-amylase, can be inhibited to represent one of the therapeutic options used in the treatment of postprandial hyperglycemia [70]. α amylase inhibitors slow the hydrolysis of starch, reducing the rate of glucose absorption and helping to maintain the blood sugar levels. A starch hydrolysis assay was conducted to determine the inhibitory effect of *W. coagulans* extract against the standard drug, acarbose, as shown in Fig 3 (c). The objective of this study was to determine the zone of inhibition and the corresponding percentage inhibition of starch degradation. At a concentration of 30 μg/mL, acarbose produced a zone of inhibition measuring 11 mm, corresponding to 69.44% inhibition of α-amylase activity as shown in Table 3. In comparison, *W. coagulans* extract at the same concentration yielded a larger inhibition zone of 15 mm but showed a lower calculated percentage of inhibition of 58.33%. Despite the results of less inhibition by *W. coagulans* compared to acarbose, the extract in question had potential to be an effective α-amylase inhibitor. The t-test was also used to compare the antidiabetic efficacy of the two treatments and the p-value was 0.000056 which showed a statistically significant difference, and thus it was established that the *W. coagulans* had a significant antidiabetic activity. These findings indicate that *W. coagulans* extract can be a potential and useful natural substitute in the management of diabetes by regulating the digestion of starch and the regulation of the postprandial glycemia. The fact that both the standard drug and the extract showed an increase in inhibition with concentration means that dosage is important in attaining effective amylase inhibition.

**3.2.2. Enzyme kinetics.** *In vitro* enzyme kinetics examination of *W. coagulans* extract has exhibited considerable antidiabetic capacity by blocking the activity of the α-amylase under a concentration-dependent pattern. As indicated

**Table 3. Starch hydrolysis assay of *W. coagulans* extract and acarbose (30 µg/mL). The assay was performed in triplicate biological replicates, each with triplicate technical replicates. Data are presented as mean±standard deviation (SD).**

| Solution (30 µg/mL) | Zone of Inhibition (mm) | % Inhibition |
|---|---|---|
| Standard (Acarbose) | 11±0.36 | 69.44 |
| *Withania coagulans* | 15±0.15 | 58.33 |

in Fig 2 the extract had an inhibition of 64.66% at 10 µg/mL and this value gradually risen to 70.26 and 75.8% at 20 and 30 µg/mL respectively. These values compared with the standard drug, acarbose that exhibited an inhibition of 65.33%, 72.66% and 80.86% and inhibition of the same at the respective concentrations. Despite the fact that acarbose demonstrated a little more inhibitory activity, the extract was still doing an excellent job, especially at low concentration, where the variation of the extract and the standard was negligible. Such biochemical findings are closely consistent with molecular docking-based findings where both Ligand-C1 and Caryophyllene oxide (Ligand-C2) were found to have high predicted binding affinities (−7.5 and −6.3 kcal/mol, respectively) with 5U3A-crystallized 5U3A. The percentages of inhibition were observed at the concentration of 30 µg/mL (75.8% with extract and 80.86% with acarbose) as supporting

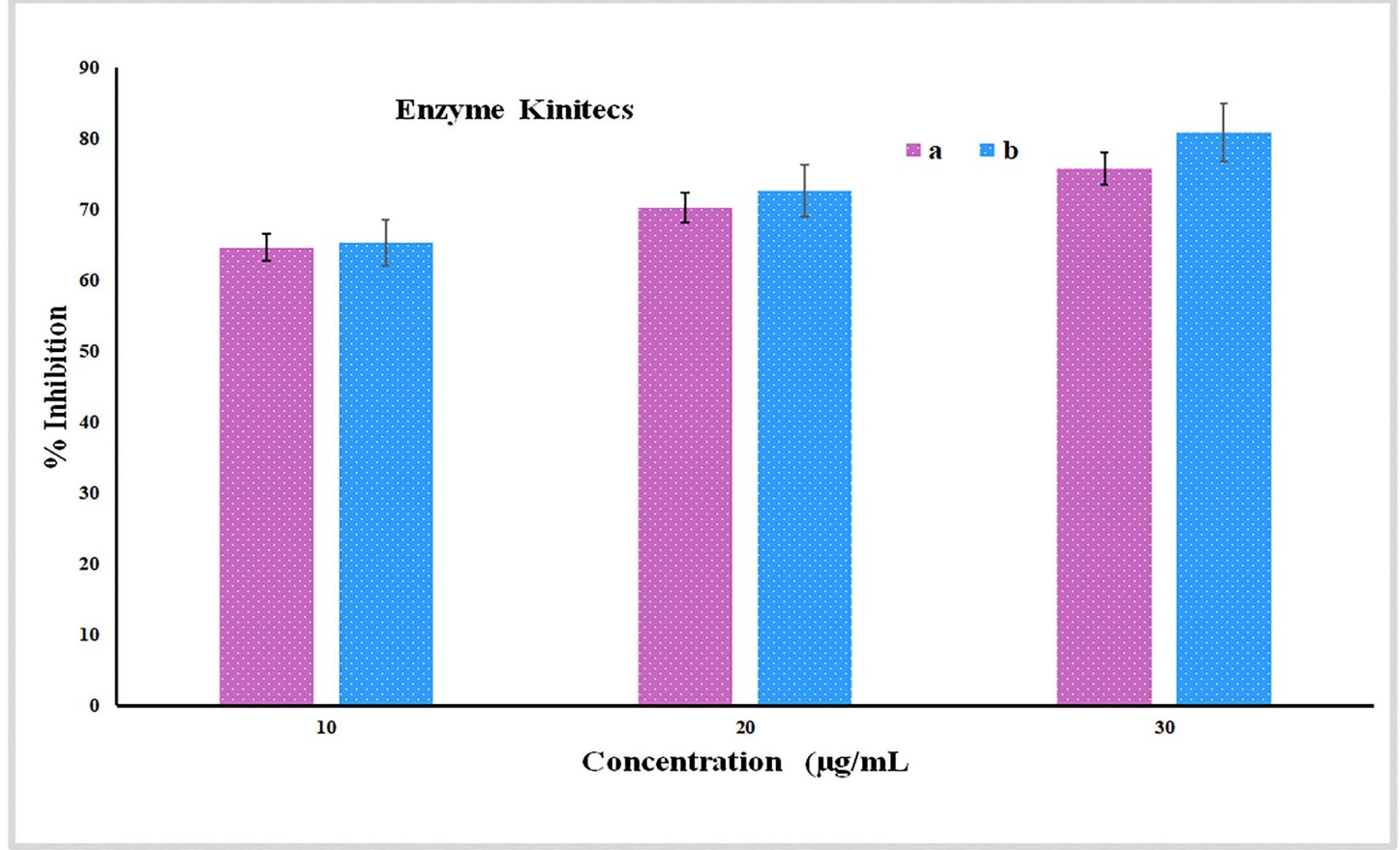

**Fig 2. Enzyme kinetics assay showing the percentage inhibition ofα-amylase by *W. coagulans* extract (a) and standard acarbose (b) at different concentrations (10–30 µg/mL).** The assay was performed in triplicate biological replicates, each with triplicate technical replicates. Data are presented as mean±standard deviation (SD).

the *in silico* predictions, indicating the possibility that these compounds may be involved in the antidiabetic action of the extract through effective inhibition of the AMY2A active site.

### 3.3. Antibacterial assay

The emergence of resistance to the traditional antibiotics in bacteria has become a significant issue in the world health field, prompting scientists to find other antimicrobial agents [71]. Although standard antibiotics are still effective, their extensive use has increased the emergence of resistant strains [72]. In this respect, interest has shifted to plant-based solutions because of their high concentration of bioactive metabolites, including alkaloids, flavonoids, tannins, and phenolic compounds, which have wide-spectrum antibacterial activity [73]. *W. coagulans* is a traditional medicine that is used in several therapeutic models. Its antimicrobial effect was tested as a comparison with an antimicrobial agent *Salmonella Typhi* (ATCC 6539) and *Escherichia coli* (ATCC 25922).

According to the results (Figs 3a, b), the standard antibiotic (sample a) yielded inhibition zones of 27 mm and 23 mm with *S. Typhi* and *E. coli*, respectively, confirming its high level of antibacterial activity. Conversely, *W. coagulans* sample b had moderate activity, with a 23 mm and 15 mm zone against *S. Typhi* and *E. coli*, respectively. The inhibitory zones at a greater concentration of 30 µg/mL (sample c) were 25 mm and 20 mm, respectively (Fig 4). These findings show that *W. coagulans* extract is concentration dependent in its ability to cause antibacterial effect, which is even higher against *S. Typhi* than *E. coli.* As it is less effective than regular antibiotics, the inhibition effect is constant, which reiterates its potential for use as a natural antibacterial agent. *W. coagulans* extract was tested against erythromycin and compared to the antimicrobial activity using CLSI (M100) guidelines [26]. Concentration-dependent effect was observed with the following inhibition zone at 30 ug/mL, 25 mm (*S. Typhi*) and 20 mm (*E. coli*) compared to 27 mm and 23 mm, respectively, for erythromycin. These findings imply moderate antimicrobial efficacy, especially against *S. Typhi*, in favor of the prospect of extracts as a natural antimicrobial agent.

The antibacterial effect of *W. coagulans* is believed to be due to the synergistic effects of its phytoconstituents. GC–MS analysis of *W. coagulans* using GC35Msp has identified a number of bioactive phytochemicals (phenolic derivatives, 2-methoxy-4-vinylphenol, benzoic acid, 3-methoxybenzyl alcohol, and 4-hydroxy-3- methylacetophenone), terpenoid

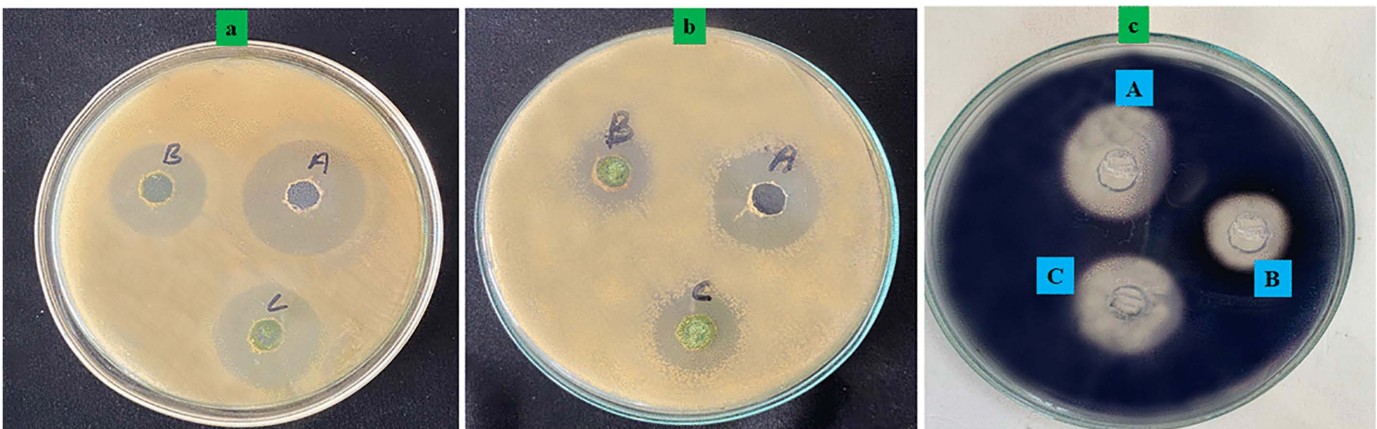

**Fig 3. Antibacterial and starch hydrolysis assays of *W. coagulans* stem extract. (a)** Zone of inhibition against *Salmonella Typhi* (ATCC 6539) and (b) against *Escherichia coli* (ATCC 25922). In both bacterial plates, A = positive control (standard antibiotic), B = *W. coagulans* extract (20 µg/mL), and C = *W. coagulans* extract (30 µg/mL). **(c)** Starch hydrolysis assay showing inhibition of α-amylase activity: A = α-amylase enzyme only (control), B = standard antidiabetic drug (acarbose), C = *W. coagulans* extract. Antibacterial assays were performed with three biological replicates, each with three technical replicates. Data are presented as mean ± standard deviation (SD).

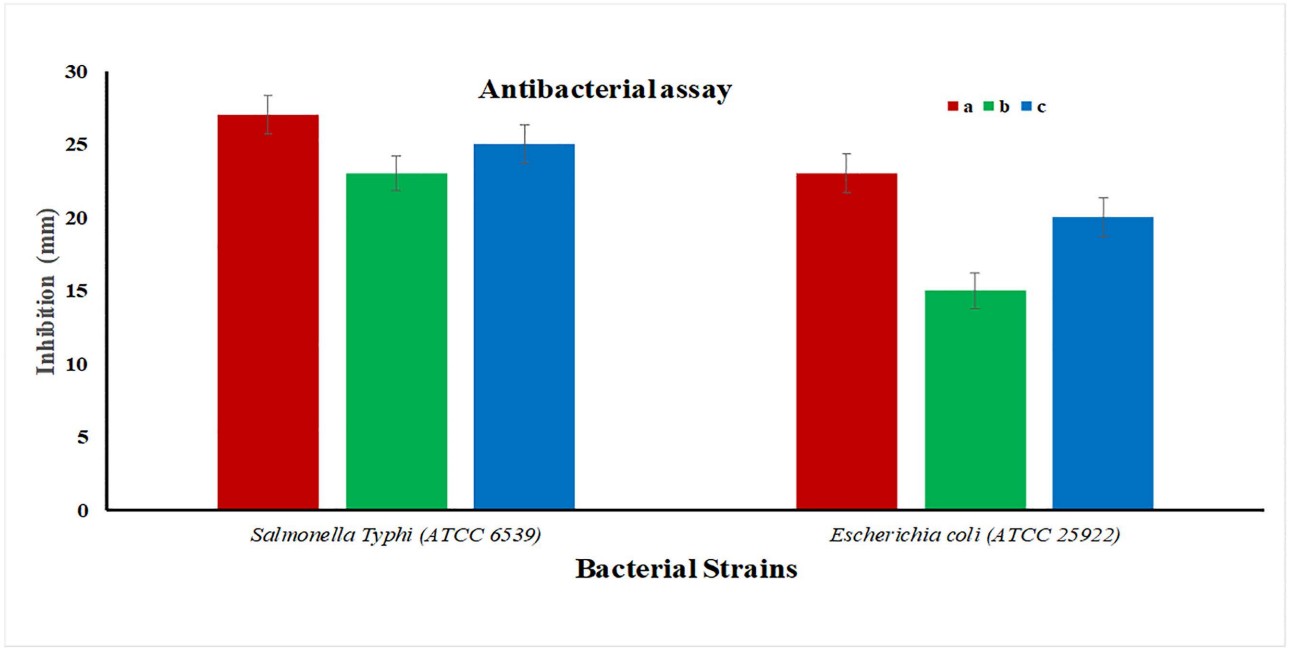

**Fig 4. Antibacterial activity of *W. coagulans* stem extract against *Salmonella Typhi* (ATCC 6539) and *Escherichia coli* (ATCC 25922).** Bars represent inhibition zones for a = standard antibiotic, b = *W. coagulans* extract (20 µg/mL), and c = *W. coagulans* extract (30 µg/mL). Data are presented as mean ± standard deviation (SD).

compounds (caryophyllene oxide, carane compounds), and fatty acids (hexadecanoic acid and its methyl ester). These compounds are synergistic and can affect bacteria through several mechanisms [74]. Lipophilic fatty acids and terpenoids intercalate into bacterial membranes, inducing lipid bilayer disruption and heightened permeability, resulting in the leakage of ions and metabolites [75]. Phenolic compounds also assist in binding to bacterial proteins and enzymes by hydrogen bonding and π – interaction leading to conformational shifts and inhibition of catalytic activity, whereas benzoic acid also acidifies the cytoplasm thereby inhibiting enzyme activity. In addition, epoxide group molecules, including caryophyllene oxide, can alkylate nucleophilic amino acid residues in vital metabolic activities, thus disrupting critical metabolic processes. This is also chelated with some phenolic acids, which stabilize divalent cations that stabilize the outer membrane of gram-negative bacteria, causing destabilization of lipopolysaccharide layers [76]. Collectively, these interactions account for the inhibition of *Salmonella Typhi* and *Escherichia coli*, supporting the antibacterial potential of *W. coagulan,* as evidenced by the agar well diffusion assay.

The aqueous stem extract of *W. coagulans* inhibited the growth of *S. Typhi* and *E. coli,* as observed in the MIC and MBC assays. The MICs were 20 µg/mL for *S. Typhi* and 30 µg/mL for *E. coli* (Table 4). For both *S. Typhi* and *E. coli*, the

**Table 4. MIC and MBC values of *W. coagulans* aqueous extract and positive control erythromycin. Data are presented as mean ± standard deviation (SD).**

| Bacterial Strain | MIC (µg/mL) | MBC (µg/mL) |
|---|---|---|
| *Salmonella Typhi* | Erythromycin: 10 ± 1.2 | Erythromycin: 20 ± 1.0 |
| | *W. coagulans*: 20 ± 1.3 | *W. coagulans*: 30 ± 1.3 |
| *Escherichia coli* | Erythromycin: 20 ± 1.1 | Erythromycin: 20 ± 1.2 |
| | *W. coagulans*: 30 ± 1.2 | *W. coagulans*: 50 ± 1.3 |

MBC values were 30 µg/mL and 50 µg/mL respectively, which implied that bactericidal activities could be exhibited at higher concentrations. The MIC/MBC indicates a bactericidal effect, which may be attributed to the activity of phytochemicals such as caryophyllene oxide, benzoic acid, and phenolic compounds that were detected through GC–MS and are known to interfere with bacterial membranes and metabolic pathways.

### 3.4. DPPH assay

The DPPH free radical scavenging test revealed that the plant extract (PE) had a concentration dependent antioxidant activity with RSA values ranging between 27.5% −57.3% in the presence of 100−500 µg/mL of the extract (Fig 5). Though this was lower than that of standard ascorbic acid (63.88% at 500 µg/mL), the extract thus had very significant radical-scavenging potential. These results were supported by a GC-MS 2 -methoxy-4-vinylphenol, benzoic acid, 3-methoxybenzyl alcohol, 4-hydroxy-3-methylacetophenone and other antioxidant phytoconstituents including phenolic and aromatic compounds were also found. These compounds are distinguished by their ability to donate hydrogen atoms/electrons to stabilize reactive species and quench free radicals. Moreover, the antioxidant activity is also observed to be caused in part by the fatty acids, such as hexadecanoic acid, and terpenoids, such as caryophyllene oxide, which stabilize

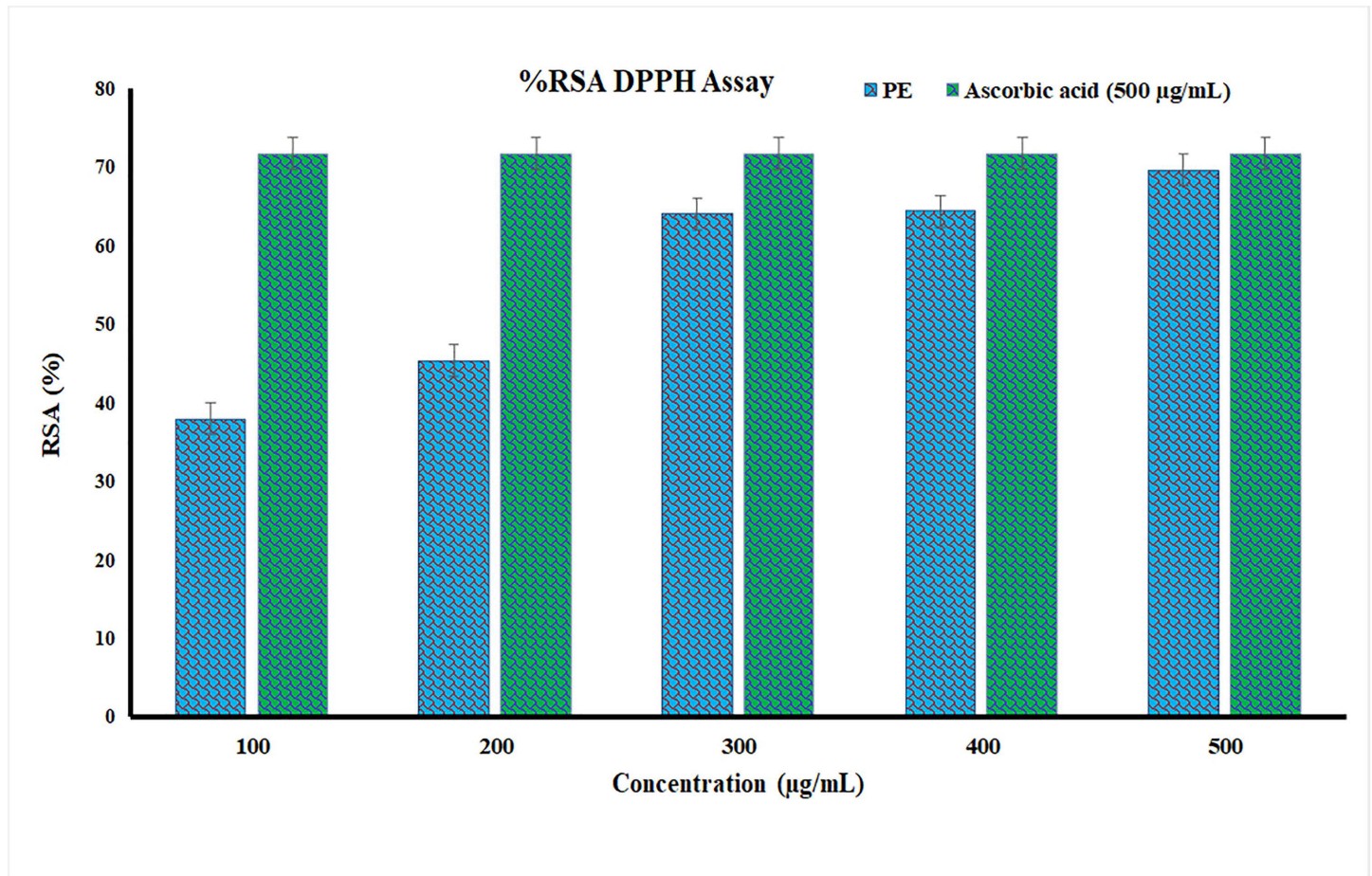

**Fig 5. DPPH radical scavenging activity (%RSA) of *W. coagulans* aqueous extract (PE) at different concentrations (100–500 µg/mL) compared with standard ascorbic acid (500 µg/mL).** Values are expressed as mean ± SD (n = 3).

the membrane of the body and regulate oxidative stress. The progressive increase in radical scavenging with concentration indicates the cumulative and synergistic effects of such phytochemicals, indicating the possible use of the extract as a natural source of antioxidants, albeit at lower concentrations than ascorbic acid.

### 3.5. *In Silico* pharmacokinetic and drug-likeness evaluation (ADMET)

To assess the therapeutic potential of the identified compounds, Absorption, Distribution, Metabolism, Excretion, and Toxicity (ADMET) of these compounds were tested in silico. S2 Table summarizes the results and provides important information regarding the drug-like properties of these compounds. Cross validation of the chosen compounds was done in pkCSM (S3 Table).

   The majority of the compounds exhibited good pharmacokinetic properties. More significantly, majority of compounds were anticipated to possess high gastrointestinal (GI) absorption which is paramount aspect of orally taken medicines. Also, some compounds were predicted to be blood-brain barrier (BBB) permeant, which indicates that they might be active in the central nervous system.

   The rule of five by Lipinski was used to determine oral bioavailability. There was nearly no violation of more than one (31 of 32) compound, which means that the oral drug candidacy is good. In general, the strong scores in bioavailability (0.55 or 0.85) of most compounds support the idea that they are good therapeutic leads. Some compounds had similar predictions with regard to ADMET despite this similarity indicates common physicochemical properties and also the constraints of the QSAR-based systems, which tend to provide similar results with structurally different phytochemicals too. Hence, such *in silico* findings may thus be considered as tentative and additional *in vivo* experiments are necessary to determine the real pharmacokinetics differences.

### 3.6. Molecular docking analysis

To elucidate the molecular mechanisms underlying the antibacterial, antidiabetic, and antioxidant activities of *W. coagulans*, molecular docking studies were performed on four key protein targets. Caryophyllene oxide named as Ligand-$C_2$ (−7.2, −7.4, −6.3, −5.6) and 2, 2- Dimethyl-3 (…)-oxirane (−6.9, −7.6, −7.5, −6.3) named as Ligand-$C_1$, were found to be the compounds with the best binding affinities to all the four targets, i.e., AMY2A (antidiabetic, PDB ID: 5U3A), HipBST (E. coli, antibacterial, PDB ID: 7AB4) and another high binding ligand was the 1-Methylene-2b-hydroxymethyl-…-cyclohexane (–6.8 to −5.5) and 4-Hydroxy-3-methylacetophenone (−6.1 to −5.2). In comparison, the interactions of compounds such as Dimefox (–3.7 to −4.1) and Dimethyl N, N-dimethylphosphoramidate (–3.5 to −4.0) were the weakest. Overall, the findings identified caryophyllene oxide and the oxirane derivative as the top potential candidates with predicted multi-target antibacterial, antidiabetic, and antioxidant properties. The binding affinities (in kcal/mol) of all 32 compounds are presented in S4 Table. A negative and lower result indicates a more stable and strong binding interaction.

   Docking analysis indicated that the binding affinities of Ligand-$C_2$ and Ligand-$C_1$ were closely related to their potential to develop stable complexes owing to a mixture of both hydrophobic and polar interactions in the active sites of their targets. Their representation of the 2D and 3D profiles of interactions revealed that both ligands were good fillers of the binding pockets and, therefore, interacted in a combination of van der Waals contacts along with alkyl and 3, 5-hydrogen contacts and the important hydrogen bonds. These antagonizing forces paralyzed the ligands but also optimized their predicted inhibitory properties, leading to the forefront of their potential use as bioactive contenders.

   The binding of *S.typhi* protein, OmpF (PDB ID: 4KR4) porin suggested a physical blockade-based antibacterial activity. Caryophyllene oxide reacted with the lining in the pore, such as GLU116, GLY120, and ARG130, mainly through van der Waals contacts but also established a π-alkyl bond with TYR112. These interactions should orient the molecule such that it partly hinders entry into the channel. In contrast, the Ligand-$C_1$ structure was particularly long and thus ideal for completely sealing the pores. It had an extended aliphatic chain allowing large van der Waals and alkyl interactions along the

channel, and a traditional hydrogen bond with ARG130 strongly anchored the ligand. This binding pattern showed a more complete steric blockade, which explains its strong predicted antibacterial activity (Fig 6a).

The stabilization of the ligand in the *E. coli* bacterial protein HipBST (PDB ID: 7AB4) was conditional on the fine balance between nonpolar and essential polar contacts. Caryophyllene oxide established van der Waals and alkyl contacts with GLU63, ASN88, and PRO62, and ILE84 and LEU93. Importantly, one hydrogen bond with GLN66 was oriented and stabilized in the binding site. Ligand-$C_1$ became part of a more detailed binding scenario, where a larger repertoire of hydrophobic interactions and one of the hydrogen bonds with LYS64 were established. This polar interaction with such a wide hydrophobic anchoring interaction and the consequent high affinity that made it possible to induce the high affinity of the compound to interfere with bacterial processes mediated by HipBST (Fig 6b).

For the anti-diabetic protein α-amylase (PDB ID: 5U3A), hydrophobic interactions in the catalytic pocket were the main stabilizing interactions that bound the substrate and prevented its entry into the catalytic site. Ligand-$C_2$ was strongly

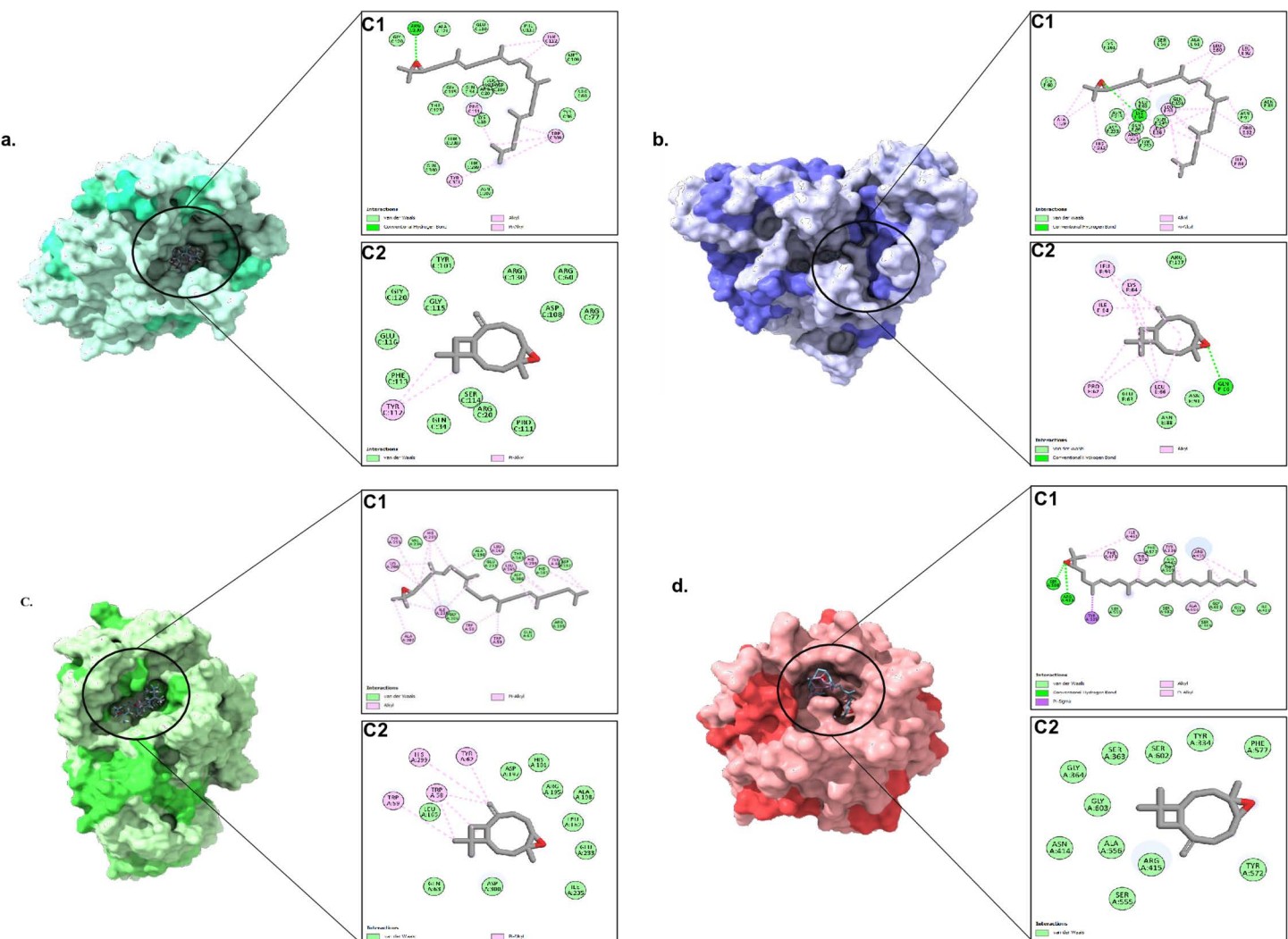

**Fig 6. Molecular docking interactions of Caryophyllene oxide and Ligand-$C_1$ with (a)** *S. typhi* OmpF porin protein (antibacterial), (b) *E. coli* HipBST protein (antibacterial), (c) *Homo sapiens* α-amylase AMY2A (PDB ID: 5U3A, antidiabetic), (d) *Homo sapiens* KEAP1 (PDB ID: 7Q6S, antioxidant/antidiabetic).

bound by residues LEU165, ASP300, and GLU233 via van der Waals contacts and enhanced its localization by aromatic residues containing TRP58, TRP59, HIS299, and TYR62. In comparison, the larger and more mobile Ligand-C1 participated in a wider set of interactions, such as with residues like VAL234, GLY306, LYS200, and HIS201, and with the same aromatic residues that stabilized Ligand-C$_2$. This compound formed a dense hydrophobic network that almost completely covered the active site, explaining its high binding affinity and inhibitory capacity against AMY2A (Fig 6c).

Their interactions with the antioxidant KEAP1 (PDB ID: 7Q6S) were characterized by a more complex interplay of hydrophobic and polar repulsions, which is essential for disrupting the regulatory role of KEAP1 in the KEAP1–Nrf2 pathway. The van der Waals interactions between residues (GLY364, SER555, and ARG415) and Ligand-C$_2$, which positioned it in the pocket, were further enhanced by cationic interactions with aromatic residues (TYR334 and PHE577). In contrast, Ligand-C$_1$ exhibited stronger and more diverse interactions. In addition to many van der Waals and alkyl interactions with residues such as TYR572 and ARG415, it also established two standard hydrogen bonds with SER508 and ARG483. A π-sigma interaction with TYR525 further stabilized the product, and a complex of three strong interactions was formed, which explains the extremely high binding score of the product and implies that it can be used as an effective KEAP1 inhibitor (Fig 6d).

### 3.7. Density functional theory (DFT) analysis

To gain insights into their electronic structure and reactivity, this study was focused on FMOs and global chemical reactivity descriptors derived from conceptual DFT based on the Koopman theorem [77,78]. The following criteria were used to filter the compounds: ADMET characterization, bioactivity evaluation, molecular docking interactions, and binding affinities. A comparison of the DFT data for caryophyllene oxide (C$_2$) and Ligand-C$_1$ is shown in Table 5.

FMOs (HOMO and LUMO) are significant orbital molecules for characterizing the optical and electronic properties of molecules (Fig 7). The HOMO and LUMO energies explain the capacity of an individual molecule to donate and accept electrons, respectively [77]. The gap between these ΔE = E_LUMO – E_HOMO] is the energy required to cause an electronic transition between the compounds. The band gap energy of the compounds revealed that Ligand-C$_1$ possessed a smaller band gap energy (6.18 eV) than Ligand-C$_2$ (6.773 eV), indicating that it was highly reactive [79]. A lower energy gap suggests that the molecule is more easily polarized and can more readily engage in charge-transfer interactions with a biological target, making it a more suitable drug candidate [80]. This computational analysis supports the potential for a strong binding affinity to the target receptor.

The energy gap (ΔE) and quantum descriptors were determined based on the calculated energy gap (ΔE). The ionization potential (I) of Ligand-C$_2$ was found to be 6.299 eV, which is higher than that of Ligand-C1 at 5.70 eV. This indicates that Ligand-C$_2$ exhibits greater chemical inertness and enhanced stability compared to Ligand-C$_1$. [81]. The chemical stability of the compound was explained using the computed chemical hardness and softness of the compounds, whereby a hard compound is more resistant to the electronic cloud than a soft molecule [82]. It was calculated that Ligand-C$_1$ (eta = 3.09 eV)

Table 5. Comprehensive DFT analysis illustrating the electronic, energetic, and quantum parameters for the top-hit compounds.

| Phase | Ligands | Dipole moment (Debye) | HOMO (eV) | LUMO (eV) | Energy gap (ΔE_gap) | Ionization potential (eV) | Electron affinity (eV) | Electro-negativity χ (eV) | Electrochemical potential μ (eV) | Hardness η (eV) | Softness S (eV$^{-1}$) | Electro-philicity ω (eV) |
|---|---|---|---|---|---|---|---|---|---|---|---|---|
| Water | Ligand-C1 | 3.649 | −5.70 | 0.48 | 6.18 | 5.70 | −0.48 | 2.61 | −2.61 | 3.09 | 0.324 | 1.10 |
| | Caryophyllene oxide | 3.5653 | −6.299 | 0.475 | 6.773 | 6.299 | −0.475 | 2.912 | −2.912 | 3.387 | 0.295 | 1.25 |
| Gas | Ligand-C1 | 3.0016 | −6.19 | 0.017 | 6.21 | 6.19 | 0.017 | 3.10 | −3.10 | 3.10 | 0.32 | 1.55 |
| | Caryophyllene oxide | 2.7918 | −6.482 | 0.312 | 6.794 | 6.482 | −0.312 | 3.085 | −3.085 | 3.397 | 0.294 | 1.401 |

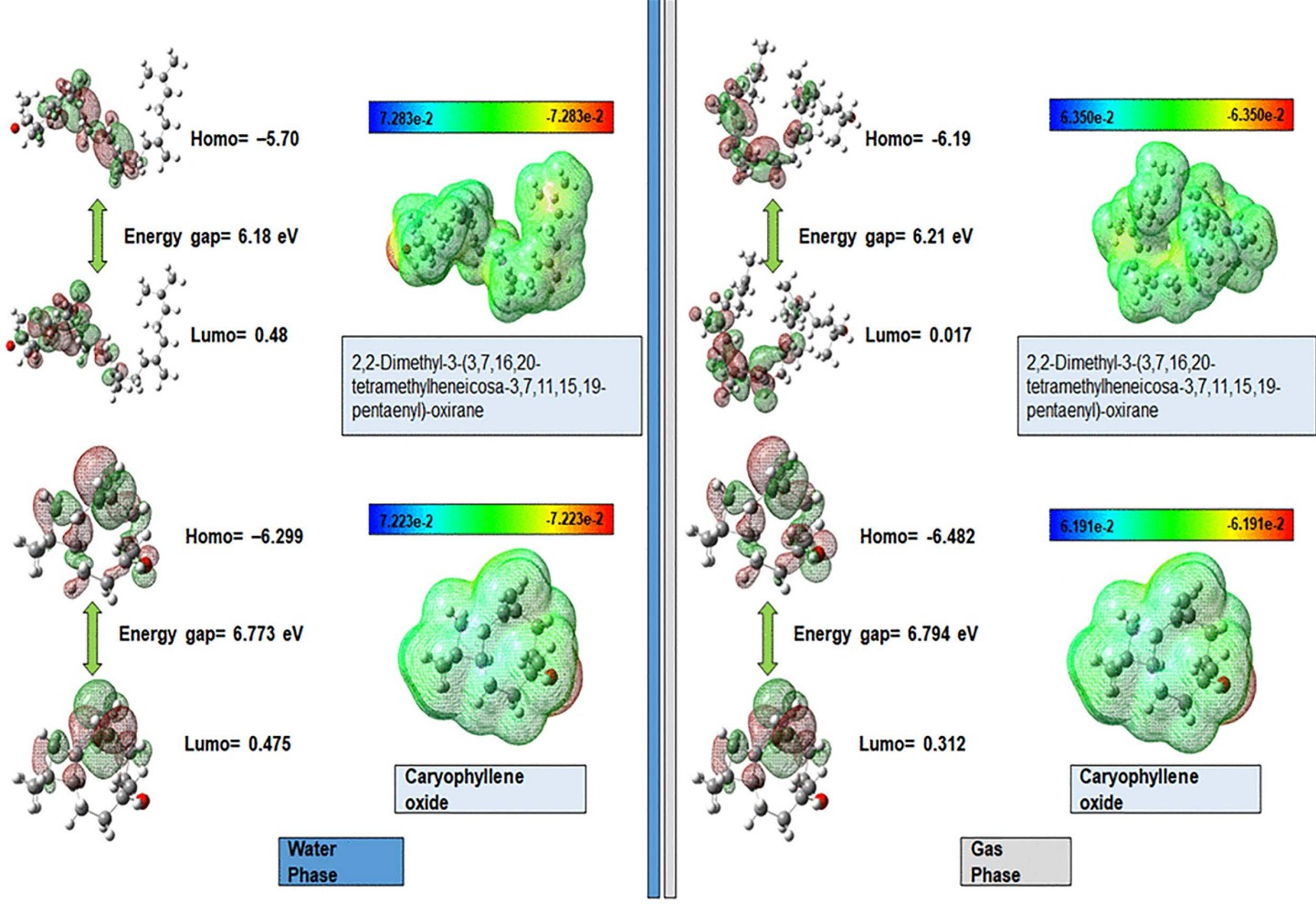

**Fig 7. Optimized structural geometry showing the Frontier Molecular Orbitals (HOMO in red/blue and LUMO in green/yellow) and Molecular Electrostatic Potential Surface (MEPS) of Ligand-C$_1$ complex and the Ligand-C$_2$ complex.** The red areas on the MEPS map indicate nucleophilic (electron-rich) regions, whereas the blue areas indicate electrophilic (electron-poor) regions.

was a softer compound than Ligand-C$_2$ (eta = 3.387 eV). Furthermore, the electrophilicity (eV) of caryophyllene oxide (1.25 eV) was slightly greater than that of Ligand-C$_1$ (1.10 eV), which is a slightly stronger electron acceptor [83].

Thus, the analysis of these reactivity descriptors builds a cohesive picture of the catalytic activity. The higher softness, lower ionization potential, and lower energy gap of Ligand-C$_1$ consistently indicates greater chemical reactivity compared to the other lead compounds. As indicated by Hussein and Azeez (2023), such reactivity profiles are often correlated with promising bioactivity, providing a strong quantum chemical basis for the superior binding affinity observed for Ligand-C$_1$ in our docking results [84].

### 3.8. Molecular electrostatic potential surface (MEPS) mapping

The MEPS shows the distribution of charge over the entire molecule and helps predict the potential sites of interaction with biomolecular targets. Fig 7 MEPS mapping is the charge distribution of compounds that were top-hit in a gradient color-coded progression of red to blue, which means that the energy distribution of a gradient between negative

(nucleophilic) and positive (electrophilic) electrostatic potential. Geometries were created and visualized in GaussView 5.0.8. [77]. Both caryophyllene oxide and Ligand-C1 displayed a distinct nucleophilic region (red) centered on the oxygen atom of the epoxide bond. This implies that the epoxide group is the most favorable location for hydrogen bond reception, which coincides with the main polar interactions in the molecular docking display. The remaining hydrocarbon structures had a neutral potential (green), indicating that they were lipophilic and thus allowed hydrophobic interactions in the binding pocket.

### 3.9. Molecular simulation

The conformational stability of the protein-ligand complexes was assessed by determining the C + RMSD of the protein backbones throughout the 50 ns simulation path (Figs 8a-d). A level plateau in the RMSD profile after the early equilibration phase is an overall indication of structural stabilization and effective formation of a stable protein-ligand complex. In all systems, the RMSD values showed that the protein backbones were intact throughout the simulations, and there were different stability patterns for the two ligands.

In the case of the antibacterial targets, both ligands would establish a stable complex with *S. typhi* protein OmpF (PDB: 4KR4), but the $C_2$ complex could reach equilibrium at a smaller RMSD of approximately 0.8 nm than that of the ligand-$C_1$ complex, which was approximately 1.3 nm. In 7AB4, the two ligands exhibited a stability, with fluctuations restricted to a reasonable range of 0.5–0.8 nm, indicating that neither ligand was a major structural error.

Another ligand-dependent stability profile was observed for the anti-diabetic protein, α-amylase (PDB ID: 5U3A). The system bound to Ligand-$C_2$ stabilized quickly at a low RMSD of approximately 0.8 nm, suggesting only minor rearrangements in the protein conformation. In comparison, the Ligand-$C_1$ complex equilibrated at a higher RMSD of~1.5 nm, which

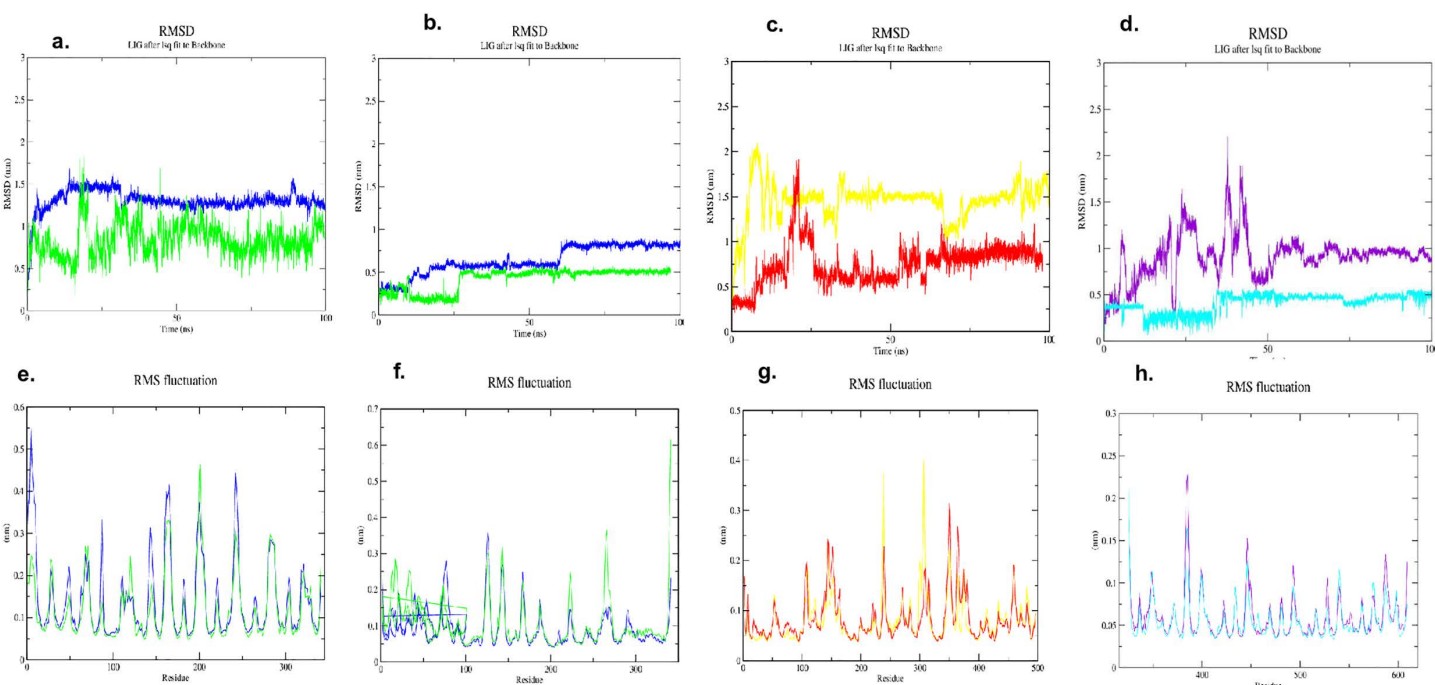

**Fig 8. Structural stability and flexibility analysis, RMSD (Cα atoms of the protein backbone over time) and RMSF (individual Cα atoms for the protein-ligand complexes) plot over 100 ns MD simulation of the (a,e) bacterial OmpF- (PDB ID: 4KR4), (b, f) HipBST- (PDB ID: 7AB4), (c, g) AMY2A- (PDB ID: 5U3A), (d, h) KEAP1- (PDB ID: 7Q6S).**

suggesting that moderate backbone rearrangements were needed to fit the ligand. Although this difference was observed, both complexes attained stable plateaus, indicating that they were capable of establishing stable binding interactions. High stability was identified as the most prominent feature of the antioxidant protein-KEAP1 (PDB ID: 7Q6S). Here, the Ligand-$C_1$ complex was stabilized below 1.2 nm compared to the Ligand-$C_2$ complex, which was further stabilized at a very low RMSD plateau of 0.5 nm throughout the simulation. This implies that Ligand-$C_2$ can bind to its protein and, stabilize the protein in a highly rigid state, thereby minimizing structural drift.

Overall, the RMSD analysis revealed that the equilibria of all simulated systems were stable, which resulted in relevant docking predictions. Remarkably, however, Ligand-$C_2$ could always stabilize smaller RMSD values than the antibacterial, antidiabetic, and antioxidant targets, which is indicative of the capacity to stabilize the protein backbone, which is better in Ligand-$C_2$. Among them, the Ligand-$C_2$-7Q6S interaction was the most stable complex, which refines the prospects of Ligand-$C_2$ as a strong stabilizer in therapeutic practice.

The Root Mean Square Fluctuation (RMSF) of all C-alpha atoms was computed as a measure of the effect of ligand-binding on local protein dynamics during the 100 ns trajectory (Fig 8e-h). The RMSF profile differentiates between rigid areas of the protein and more pliable domains.

The characteristic pattern of fluctuations in all complexes was as follows: residues of stable secondary structures (α-helices and β-sheets) were rigid with low RMSF values (usually less than 0.2 nm). Conversely, more motile movements were observed in intrinsically flexible loop motifs and at the N- and C-termini. The antibacterial target proteins OmpF (PDB ID: 4KR4) and HipBST (PDB ID: 7AB4) exhibited relatively lower fluctuations (< 0.3 nm) across their backbones, with few localized peaks representing surface loops (Figs 8e, f). This shows that the binding of both Ligand-$C_1$ and Ligand-$C_2$ stabilized the protein structures without causing excessive flexibility.

The antidiabetic target protein -AMY2A (PDB ID: 5U3A) exhibited moderate fluctuation peaks of approximately 0.4 nm in certain loop areas, a typical feature of proteins with dynamic active sites (Fig 7g). Despite this localized mobility, the general structure was stable. Importantly, the antioxidant target protein KEAP1 (PDB ID: 7Q6S) presented the best overall RMSF profile (< 0.25 nm), indicating that the ligands of the antioxidant target provided the maximum rigidity to the protein backbone (Fig 8h).

Importantly, in all simulated systems, the amino acid residues that formed the binding pocket showed low RMSF values. This localized rigidity is a good indication that the ligands were well-anchored in the active site and that they underwent continuous interactions during the simulation, which is a good indication of the formation of a stable binding mode.

### 3.10. Structural Compactness and Integrity through Rg and SASA

The Radius of Gyration (Rg) and the solvent accessible surface area (SASA) were used to determine the extent of compactness and folding integrity of the protein-ligand complexes. Rg acts as a measure of the overall size of the protein, and the Rg values remained constant, indicating a regular fold.

All complexes showed high stability in the Rg profiles over the course of the 100 ns simulation, with no pronounced trends indicative of unfolding or aggregation (Figs 9a-d). In particular, the antibacterial complexes (4KR4 and 7AB4) varied close to 2.1–2.3 nm, and the antidiabetic complex (5U3A) was steady at approximately 2.3–2.4 nm. The antioxidant KEAP1-complex (PDB ID: 7Q6S) retained the smallest profile, and there was minimal fluctuation around a value of approximately 1.8 nm, which is inconsistent with the high rigidity of the complex in the RMSF analysis.

These observations of sustained compactness were further supported by the SASA analysis (data shown in (Figs 9e-h), which confirmed that the total solvent-exposed surface area of the proteins remained constant after equilibration. Collectively, the stable Rg and SASA profiles provide robust evidence that the ligands form stable complexes without compromising the global structural integrity or compactness of target proteins.

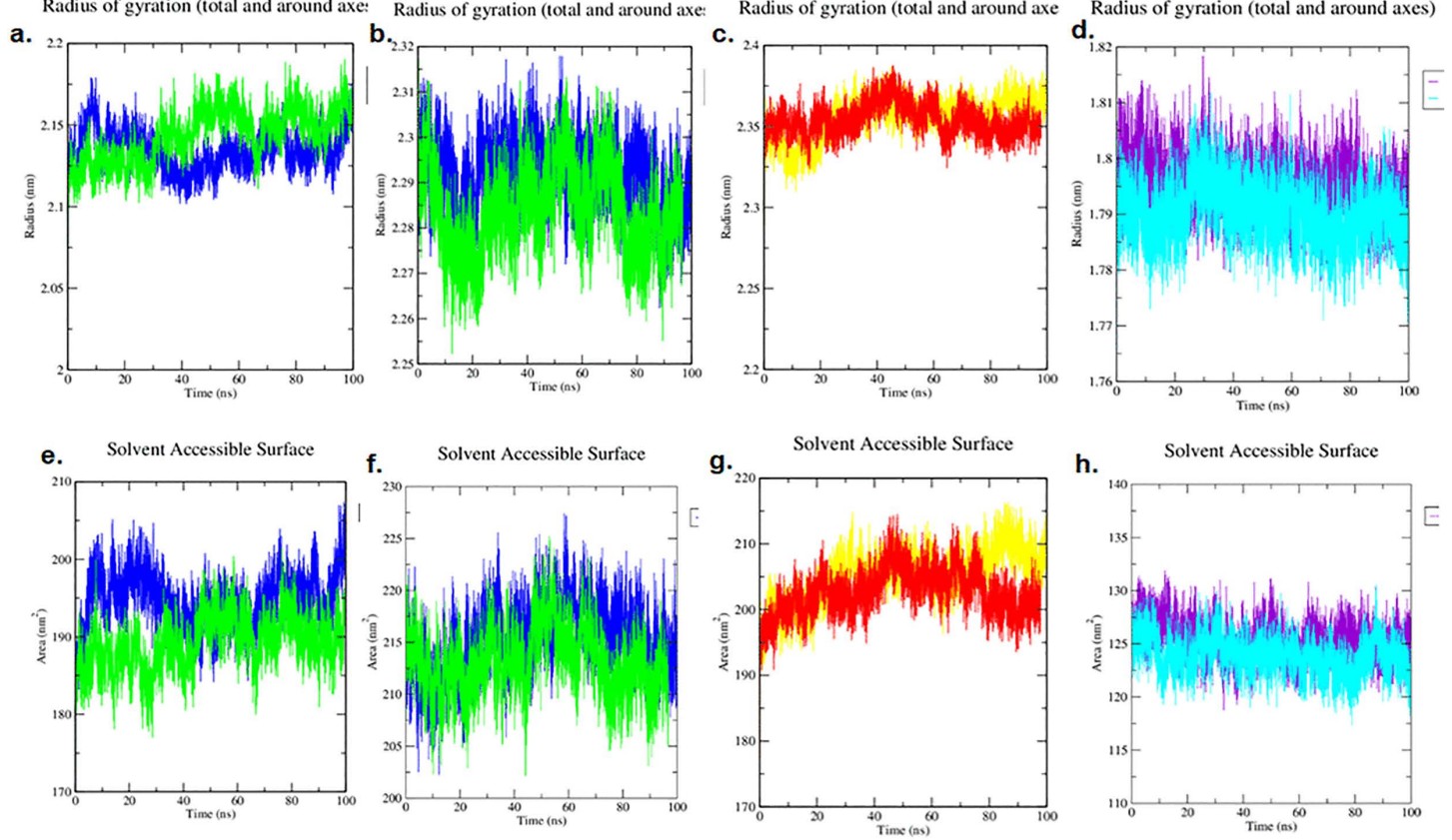

**Fig 9. Analysis of structural compactness and integrity from 100 ns MD simulations, Radius of Gyration (Rg) and Solvent Accessible Surface Area (SASA) of (a,e) bacterial OmpF- (PDB ID: 4KR4)(b, f) HipBST- (PDB ID: 7AB4), (c, g) AMY2A- (PDB ID: 5U3A) (d, h) KEAP1- (PDB ID: 7Q6S).**

### 3.11. Principal component analysis of conformational dynamics

To investigate the large-scale collective motions of the protein-ligand complexes and characterize the major conformational states sampled during the simulations, Principal Component Analysis (PCA) was performed. This analysis is based on covariance matrices derived from the fluctuations of Cα atoms, where the resulting eigenvectors (principal components, PCs) represent the dominant modes of atomic motion [85]. The C-alpha trajectory of each protein-ligand complex was projected onto the first two principal components (PC1 and PC2).

In the case of the antibacterial protein OmpF (PDB: 4KR4), the PCA plots indicated that the two ligands were sampled differently (Figs 10a-b). In the presence of Ligand-$C_1$, the protein explored an extensive conformational space and formed two major, well-separated clusters along PC1. Conversely, the complex with Ligand-$C_2$ occupied the phase space in a more tightly clustered and compact manner. Likewise, in the case of the antibacterial- HipBST (PDB ID: 7AB4), the Ligand-$C_1$ complex exhibited an uninterrupted conformational transition between states, whereas the Ligand-$C_2$ complex exhibited a more limited conformational space with increased clustering (Figs 9c-d). This indicates that the binding of Ligand-$C_2$ to the protein suppresses its collective motions, resulting in a stable and structurally conserved state of both antibacterial targets.

Significant changes in the conformational dynamics due to the two ligands were also reported by the antidiabetic protein -AMY2A (PDB ID: 5U3A) PCA (Figs 10e-f). The complex with Ligand-$C_1$ was sampled in a large conformational

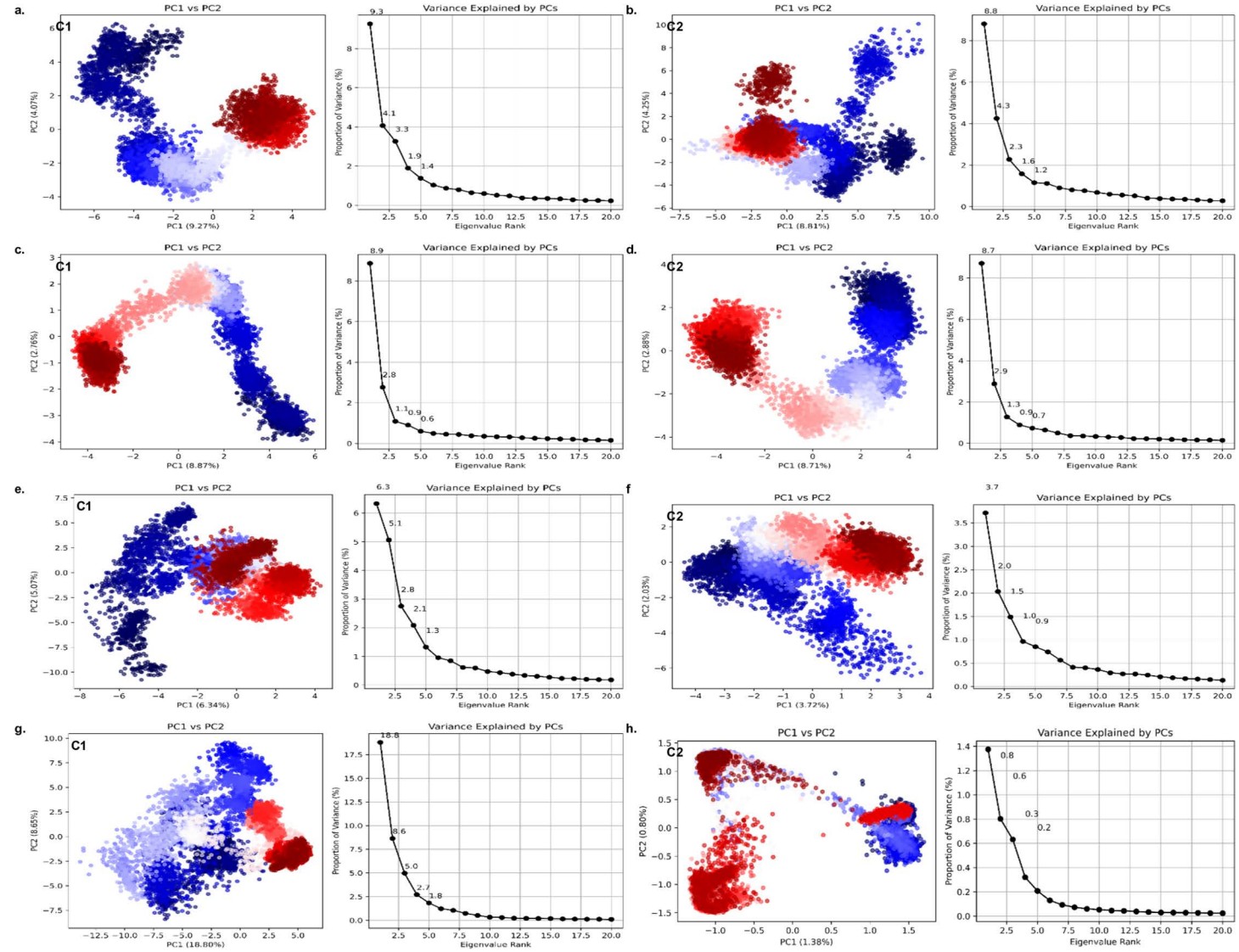

**Fig 10. Principal Component Analysis of Conformational Dynamics.** 2D projection plots of the first two principal components (PC1 and PC2) showing the conformational space sampled by the Cα atoms during the 100 ns MD simulation of $C_2$ and $C_1$ **(a,b)** OmpF (PDB: 4KR4),(c,d) HipBST (PDB ID: 7AB4), **(e,f)** AMY2A (PDB ID: 5U3A), **(g,h)** KEAP1(PDB ID: 7Q6S), respectively.

space, and clusters were observed that signified a transition between the various metastable states. This indicates that the degree of flexibility and structural reorganization of the protein is rather high. In contrast, the 5U3A-Ligand-$C_2$ complex had a strikingly smaller spread of conformations. The states sampled were restricted to a much smaller area of the phase space, and this single cluster was dense. This implies that the large-scale motions of 5U3A were mostly suppressed by Ligand-$C_2$ binding, which placed it in a more stable and less dynamic position.

The antioxidant protein KEAP1 (PDB ID: 7Q6S) showed the most dramatic stabilization effect (Figs 10g-h). Although stable, the Ligand C1 complex still visited a significant part of the conformational space, as evidenced by the broad distribution along PC1, of which 18.80% of the total variance was contributed. The 7Q6S-Ligand-$C_2$ complex, in sharp contrast, was highly rigid. The entire path of the simulation was restricted to an extremely small and localized area of the phase

space, and PC1 explained only 1.38% of the variance. This extremely low variance indicates that Ligand-$C_2$ almost completely suppresses the collective movements of the protein as whole, which imposes a very stable and conformationally restricted state. This observation is in agreement with the very low values of RMSD and RMSF for this complex.

In the summary, the Principal Component Analysis provides a coherent view of the dynamic behavior of all targets. An apparent and coherent pattern was observed in which Ligand-$C_2$ was a better conformational stabilizer than Ligand-$C_1$. For antibacterial, antidiabetic, and antioxidant targets, the production of Ligand-$C_2$ resulted in a smaller and less diverse conformational assembly. This reduced sampling of the phase space strongly suggests that Ligand-$C_2$ forms more rigid and stable complexes [86], which aligns perfectly with the observations from RMSD and RMSF analyses. These findings underscore that Ligand-$C_2$ has a remarkable capacity to induce a stable, inhibited state across multiple functionally distinct protein targets.

### 3.12. Dynamic cross-correlation matrix (DCCM) analysis

Protein regions often exhibit correlated movements that can be quantitatively assessed using dynamic cross-correlation matrix (DCCM) analysis. These motions are pivotal for understanding conformational transitions, allosteric communication, and protein stability during molecular dynamics simulations [87]. In this study, DCCM analysis of the Cα backbone atoms of each protein in complex with its ligands was performed based on the 100 ns trajectory data. The ensuing maps revealed areas of strongly correlated motion (positive red), anti-correlated motion (negative blue), and uncorrelated areas (white).

In the case of antibacterial targets, the dynamic protein effects depend on the ligands. Both Ligand-$C_1$ (Fig 11a) and Ligand-$C_2$ (Fig 10b) seemed to share the distribution of strong positive correlations, predominantly local secondary structure elements, with a small number of anti-correlated motions with the OmpF protein (PDB: 4KR4). This shows that the two ligands did not modify the domain communication inherent to the protein, resulting in significant dynamic shifts.

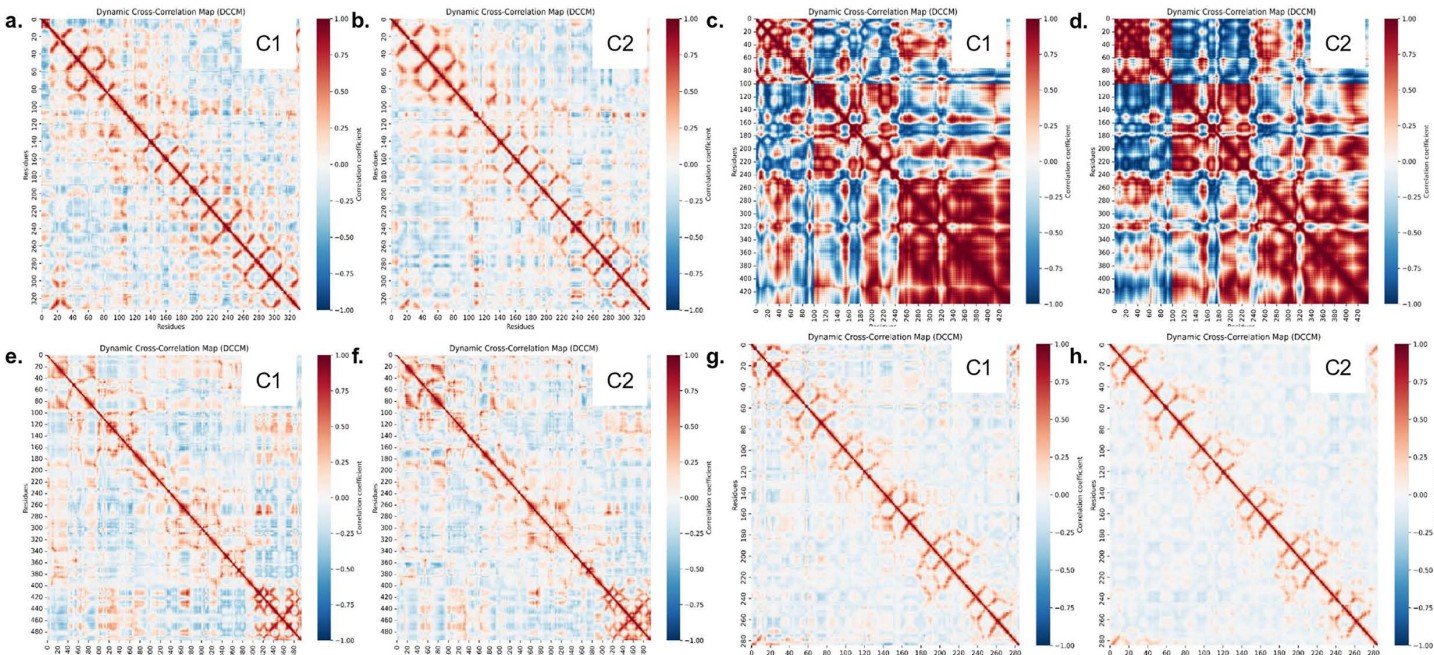

**Fig 11. Dynamic Cross-Correlation Matrix (DCCM) analysis of atomic motions.** The maps illustrate correlated (red), anti-correlated (blue), and uncorrelated (white) movements of Cα atoms during the 100 ns MD simulation for **(a,b)** OmpF (PDB: 4KR4),(c,d) HipBST (PDB ID: 7AB4) **(e,f)** AMY2A (PDB ID: 5U3A), **(g,h)** KEAP1 (PDB ID: 7Q6S) with $C_2$ and $C_1$ ligands respectively.

The action of the protein-7AB4, on the other hand, was quite different. The Ligand-C$_1$ complex ([Fig 11c]) showed high anti-correlation sites over large regions (violet blue), implying that the entire region of the protein was sliding against each other, which is a large-scale, hinge-like motion. However, the range of anti-correlated motions was greatly diminished in the Ligand-C$_2$ complex ([Fig 11d]) and greatly increased as positive correlations (red), indicating a more coordinated and coherent movement throughout the protei than in the WTn. This suggests that disruptive domain motions are reduced in the presence of Ligand-C$_2$ binding, in contrast to the presence of Ligand-C$_1$, resulting in higher native structural stability.

With the protein -AMY2A (PBD: 5U3A), a slight variation was observed in the two ligands, albeit minor. The Ligand-C$_1$ complex ([Fig 11e]) was slightly weakly correlated in the mean and possessed regions that appeared uncorrelated (white) or weakly anti-correlated (light blue), suggesting a less-structured and more flexible dynamic structure. Comparatively, the Ligand-C$_2$ complex ([Fig 11f]) showed a wide range of positive correlations with greater intensities and extents of positive correlations than Ligand-C$_1$, with larger red blocks at the diagonal and fewer anticorrelation movements. This observation indicates that Ligand-C$_2$ increases the concerted and homogeneous movement of the protein domains, which stabilizes the overall conformation.

The strongest stabilizing effect was observed on the antioxidant protein KEAP1 (PDB 7Q6S) ([Fig 11g]). Ligand-C$_1$ complex was weakly/moderately correlated with scattered patches of positive and negative motions, suggesting that different domains of the protein were independent of each other and moved independently with little or no interaction. In contrast, both the Ligand-C$_2$ complex ([Fig 11h]) demonstrated large-scale and strong positive correlations, and there were almost no anti-correlated movements. The strong, long-range red blocks predominant along the diagonal of the DCCM map demonstrate that Ligand-C$_2$ binding effectively locks the domains of the protein in a highly coordinated and rigid functional state.

Overall, the DCCM analysis supports the results of the RMSD, RMSF, and PCA, where the mechanism of stability due to the ligands is observed. For all targets, Ligand-C$_2$ persistently increased concerted positive correlations and decreased anti-correlated motions, which contributed to increased structural rigidity and stability compared with Ligand-C$_1$. This stabilization was best observed in the antioxidant protein KEAP1 (PDB ID: 7Q6S), where the binding of Ligand-C$_2$ resulted in an unusually rigid conformation [86]. These results highlight the superior ability of Ligand-C$_2$ to promote cohesive and stable protein–ligand complexes with therapeutic potential.

### 3.13. Binding free energy calculation using MM/GBSA

Although molecular docking provides valuable initial predictions, its scoring functions often lack the precision required for definitive ranking of compounds [88]. To achieve a more accurate and reliable estimation of the binding affinities, the Molecular Mechanics/Generalized Born Surface Area (MM/GBSA) method was employed.

By incorporating solvation contributions and conformational averaging from the stable MD trajectory, this approach yielded more robust free-energy estimates for ligand ranking. The MM/GBSA analysis demonstrated a clear and reproducible preference across all four therapeutic targets, with 2,2-Dimethyl-3-(…)-oxirane consistently showing stronger binding interactions than caryophyllene oxide. The higher affinity of 2,2-Dimethyl-3-(…)-oxirane (ΔGTOTAL: –26.52 to –40.13 kcal/mol) relative to Caryophyllene oxide (ΔGTOTAL: –11.74 to –22.65 kcal/mol) underscores its potential as a superior lead candidate, as shown in [Fig 12] and [Table 6].

### 4. Discussion

The plant *W. coagulans* is enriched with a diverse array of phytochemicals, and its wide-ranging traditional applications suggest that its chemically active constituents may contribute to significant antidiabetic, antibacterial, and antioxidant effects [89]. This implies that the complexity of phytochemical character of crude extract of plant is what causes the therapeutic effect. The key objective of the research was to examine these phytochemical constituents and further examine their antidiabetic, antioxidant and antibacterial [90] capabilities of the *W. coagulans* stem extracts. Gas Chromatography-Mass Spectrometry (GC-MS) was used to identify the specific phytochemicals present in the aqueous extract, and it

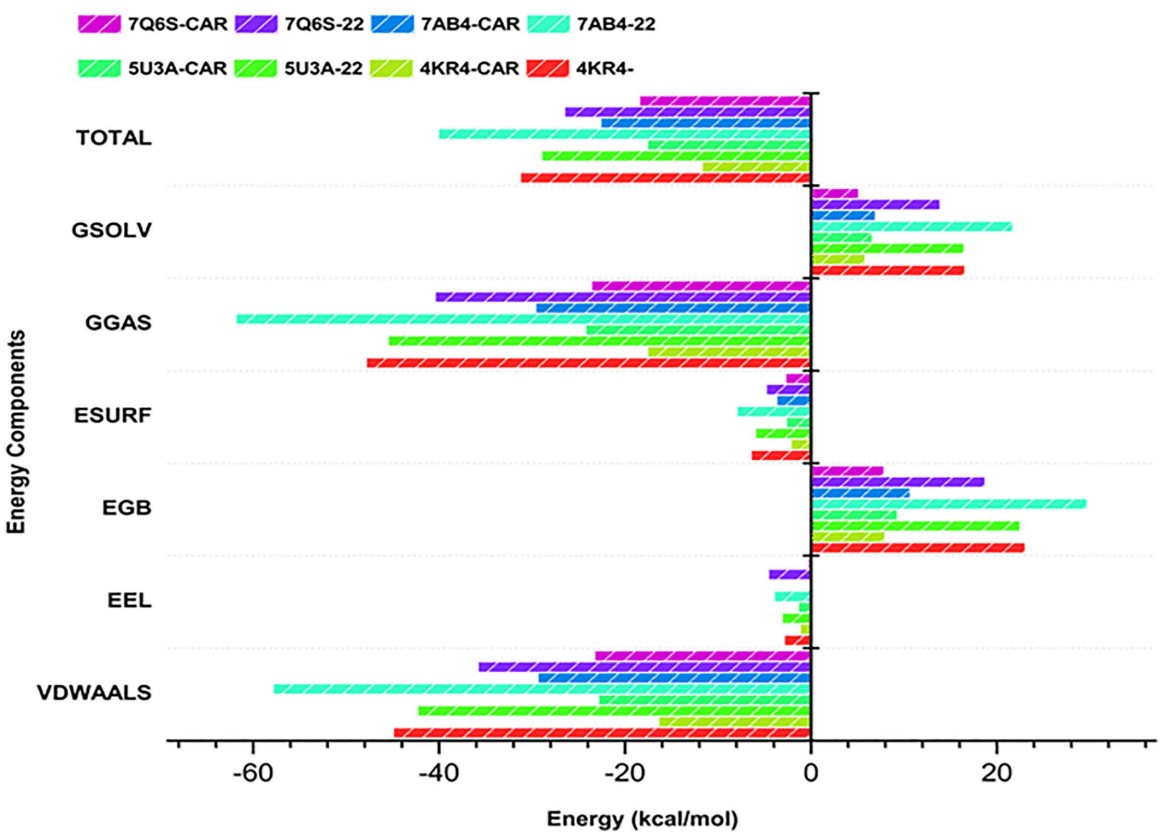

**Fig 12. Binding Free Energy Calculation using MM/GBSA** The bar chart shows the total binding free energy (ΔGTOTAL in kcal/mol) for Ligand-$C_1$ and Ligand- $C_2$ with each of the four target proteins.

**Table 6. MM/GBSA binding free energy components (kcal/mol) for each protein–ligand complex. Values are mean±SD from 50 ns of the MD simulations. More negative ΔTOTAL values indicate a stronger binding affinity.**

| Energy Component (kcal/mol) | 4KR4–C1 | 4KR4-C2 | 5U3A-C1 | 5U3A-C2 | 7AB4–C1 | 7AB4-C2 | 7Q6S-C1 | 7Q6S-C2 |
|---|---|---|---|---|---|---|---|---|
| ΔVDWAALS | −44.97±1.95 | −16.40±1.33 | −42.33±1.94 | −22.88±1.22 | −57.88±2.18 | −29.40±1.11 | −35.84±0.20 | −23.29±1.02 |
| ΔEEL | −2.90±0.91 | −1.20±0.07 | −3.17±1.13 | −1.39±0.32 | −4.00±0.75 | −0.24±0.08 | −4.62±0.63 | −0.33±0.02 |
| ΔEGB | 23.08±0.09 | 8.02±0.11 | 22.52±0.39 | 9.34±0.06 | 29.74±0.07 | 10.72±0.06 | 18.78±0.38 | 7.89±0.15 |
| ΔESURF | −6.48±0.04 | −2.16±0.00 | −6.01±0.07 | −2.68±0.03 | −7.99±0.31 | −3.72±0.04 | −4.84±0.20 | −2.74±0.06 |
| ΔGGAS | −47.87±2.15 | −17.60±1.33 | −45.51±2.25 | −24.27±1.26 | −61.88±2.30 | −29.64±1.12 | −40.46±0.66 | −23.62±1.02 |
| ΔGSOLV | 16.60±0.10 | 5.86±0.11 | 16.51±0.39 | 6.65±0.07 | 21.75±0.32 | 6.99±0.07 | 13.94±0.43 | 5.15±0.16 |
| ΔTOTAL | −31.28±2.16 | −11.74±1.34 | −29.00±2.28 | −17.62±1.26 | −40.13±2.32 | −22.65±1.12 | −26.52±0.79 | −18.47±1.04 |

showed that 32 different compounds, including terpenoids, phenols, and fatty acids, were present. GC-MS is suitable for the analysis of volatile compounds but is limited when used on aqueous extracts. It selectively absorbs nonpolar and volatile compounds and polar, nonvolatile compounds (including phenolics) are not bound to be efficiently absorbed. Moreover, the accuracy can be affected by thermal degradation and matrix interference. High-Resolution Mass Spectrometry (HRMS) would be useful to verify the identity of the key compounds, and the identity of the compounds could be confirmed by co-injection with standards. The chemical richness profile we found is in agreement with previous research, which reported the complex structure of this plant.

The major global health problem that affects worldwide healthcare systems is antibiotic resistance [91,92]. The emergence and subsequent dissemination of multidrug-resistant pathogens have posed significant problems for traditional antibiotic treatment and increased the need for new antimicrobial agents. Medicinal plants that store numerous bioactive compounds that are known to have therapeutic potential have been explored severally in this regard [93]. The present study revealed that the aqueous extract of *W. coagulans* has strong antibacterial activity against all the cultivated strains, such as *S. aureus* and *E. coli.* Our findings coincide with those of other reports that have also established the broad-spectrum antimicrobial properties of *W. coagulans* extracts [94].

To test the antidiabetic performance of the *W. coagulans* extract, an in vitro cell-free assay of alpha-amylase inhibition was conducted. The extract was also found to have a good α-amylase inhibition with an inhibition zone of 15 mm and 58.33% inhibition just like the standard drug acarbose which inhibited 69.44%. Blockage of major digestive enzymes of carbohydrates is a well-known approach to the treatment of postprandial hyperglycemia. Contrary to previous studies that have validated the hypoglycemic effects of *W. coagulans* [16], our observations affirm the use of the plant in the management of the disease through traditional use in maintaining normal glucose levels. Its antioxidant effect, as indicated by the DPPH assay, is also an addition to its therapeutic value because one of the major drivers of diabetic complications is oxidative stress. The good correlation between the calculated binding affinities and measured α-amylase inhibition is evidence of the mechanistic value of the top-scoring compounds. Its crowded hydrophobic contacts and hydrogen bonding anchor in the active site of AMY2A, Ligand-C1 was found to replicate the inhibitory effect of the conventional drug acarbose. Moreover, MD simulations confirmed the structural stability of the complex, and DFT analysis suggested high chemical reactivity (low ΔE_gap), which may facilitate electron transfer or binding. These multi-level computational results reinforce the biochemical performance of the extract, suggesting that the terpenoid-rich profile of *W. coagulans* contributes directly to its enzyme-inhibitory action.

RMSD analysis was used to assess the global conformational stability of the protein-ligand systems, as it is a well-established measure of backbone equilibration during molecular dynamics simulations [95]. Overall, smaller RMSD values indicate minimal structural deviations and stable complex formation [96]. The stability of the protein-ligand complexes was verified by complementary RMSF profiles, where fluctuations were mostly kept below 0.3 nm in the case of the antibacterial targets, around 0.4 nm in the case of the antidiabetic target, and below 0.25 nm in the case of the antioxidant target. Notably, the binding-site residues showed small variations (<0.2 nm), which confirmed the strong ligand binding and increase in structural rigidity, especially in the complexes with Ligand-C$_2$ [97].

The second key objective of this study was to explore the potential therapeutic constituents present in the aqueous extract of *W. coagulans*, with the aim of identifying possible lead phytochemicals for future drug development. To achieve this, the drug-likeness of the 32 phytochemicals was assessed using ADMET analysis. Most of these compounds, such as caryophyllene oxide and 2,2-dimethyl-3-(…)-oxirane, exhibited favorable pharmacokinetic properties, including high predicted gastro-intestinal absorption and compliance with Lipinski's rule of five. Based on these findings, the compounds were subjected to molecular docking analysis to evaluate their ability to interact with therapeutic protein targets, a computational strategy widely recognized in previous studies for predicting ligand–protein binding conformations. Molecular docking provides insights into the spatial orientation and binding compatibility of small molecules within the active sites of target proteins [86].

In this study, PyRx software was used to perform molecular docking analysis, which predicts binding affinity in terms of a docking score. A more negative score indicates a stronger interaction, which is often stabilized by forces such as hydrogen bonds, van der Waals forces, and hydrophobic contacts [98]. Among these, hydrogen bonds play a critical role in ensuring the high specificity and stability of the protein-ligand complexes [99]. Molecular docking results demonstrated that caryophyllene oxide and 2,2-Dimethyl-3-(…)-oxirane exhibited the best and most consistent binding affinities across all four therapeutic targets, including AMY2A (antidiabetic), KEAP1 (antioxidant), and the bacterial proteins HipBST and OmpF (antibacterial). This suggests that these two compounds possess multi-target binding potential. The strong binding scores of these compounds, which ranged from −6.3 to −7.6 kcal/mol, validate their potential as potent bioactive agents.

Notably, the antidiabetic activity of β-caryophyllene has been previously reported, strongly supporting our *in silico* findings [100]. Comprehensive interaction analysis confirmed their ability to form stable complexes within the active sites, making them excellent therapeutically active phytochemicals for future studies, as has been proposed in other studies based on strong docking results [99].

To place our current results within the framework of existing antimicrobial treatment and drug development, it must be noted that plant-derived compounds are becoming the subject of increasing investigations as an alternative or supplement to traditional antibiotics because of increasing antimicrobial resistance (AMR). Recent findings have shown that different phytochemicals, such as terpenoids, demonstrate antibacterial activity by acting in different ways, including cell membrane disruption and virulence factor inhibition, demonstrating the potential use of natural antimicrobials against resistant pathogens and synergistic activities with conventional drugs [101,102]. Previous research has demonstrated a strong antibacterial potential and the influence of extraction techniques and bioactive composition on efficacy, offering a rationale to compare plant extracts with existing antibiotics [103]. Although the inhibition zones and MIC/MBC of the *W. coagulans* terpenoids in our study were slightly lower than those of potent commercial antibiotics, these findings are consistent with the overall body of literature that plant-based antimicrobials may still be useful as lead scaffolds in new drug development or synergists that can lower drug dosage and delay the onset of resistance [104]. To achieve clinical translation, issues like bioavailability, pharmacokinetics, and toxicity have to be overcome by additional *in vivo* and formulation investigation, yet recent systematic reviews indicate a sense of urgency and the possibility of plant-based compounds to complement the current treatment approaches to antimicrobial agents in the future [101].

This study suggests that *W. coagulans* is enriched with bioactive phytochemicals that may contribute to its notable antibacterial, antidiabetic, and antioxidant properties. GC–MS profiling, in vitro assays, and *in silico* validation together point to caryophyllene oxide and 2,2-dimethyl-3-(…)-oxirane as promising multi-target compounds present in the extract. These findings substantiate the ethnopharmacological claims of this plant and highlight its potential as a natural source of lead molecules for drug development. Although this study demonstrates the multi-target pharmacological potential of *W. coagulans* terpenoids, experimental evaluation of cytotoxicity and *in vivo* antidiabetic effects, including histopathological assessment, was not performed due to facility and ethical limitations. Future studies will address these aspects to establish safety and validate therapeutic efficacy.

## 5. Conclusion

The present study demonstrated that the aqueous stem extract of *W. coagulans* is rich in bioactive constituents, including phenolics, terpenoids, and fatty acids, as identified by GC-MS. In vitro assays confirmed its multifunctional property. Antibacterial screening revealed inhibition zones of 15–25 mm against *Salmonella Typhi* (ATCC 6539) and *Escherichia coli* (ATCC 25922), showing activity comparable to, but slightly lower than, that of the standard antibiotic. The antidiabetic potential was validated using both the starch hydrolysis assay and enzyme kinetics studies. The extract inhibited α-amylase with 58.33% inhibition (15 mm zone at 30 µg/mL) compared to 69.44% for acarbose, and kinetic analysis further confirmed its ability to interfere with enzyme activity in a dose-dependent manner. The antioxidant capacity was supported by the DPPH assay, where radical scavenging increased from 27.5% at 100 µg/mL to 57.3% at 500 µg/mL, relative to 63.88% for ascorbic acid. These findings provide strong scientific validation for the ethnopharmacological claims surrounding *W. coagulans*. While bioactivity was confirmed using the crude extract, our *in silico* analyses identified caryophyllene oxide and 2,2-dimethyl-3-(…)-oxirane as promising lead terpenoids within the extract, exhibiting strong docking affinities, stable molecular dynamics profiles, favorable MM/GBSA free energies, and acceptable ADMET properties. Although these specific compounds are hypothesized to contribute to the observed effects, further studies involving their isolation and individual biological testing are required to confirm their exact roles. Nevertheless, the integration of GC–MS profiling, in vitro validation, and computational modelling underscores the promise of *W. coagulans* as a source for drug discovery and development.

## Acknowledgments

This study supported by Atatürk University project number FDK-2025-14849.

## Supporting information

**S1 Protocol. Stepwise Protocols S. In Silico Study – Stepwise Protocol.**
(DOCX)

**S1 Table. Phytochemical constituents identified in the aqueous stem extract of Withania coagulans by GC–MS analysis, along with their PubChem CID, molecular formula, molecular weight, canonical SMILES, and chemical structures.**
(DOCX)

**S2 Table. Drug-likeness and pharmacokinetic properties of GCMS identified compounds.**
(DOCX)

**S3 Table. Cross Validation of ADME properties of selected compounds by SwissADME and pkCSM.**
(DOCX)

**S4 Table. Docking Score of GCMS identified Compounds with targeted Proteins.**
(DOCX)

## Author contributions

**Conceptualization:** Hansa Gul, Nasir Assad, Zahida Nasreen, Muhammad Naeem-ul-Hassan.

**Data curation:** Hansa Gul, Nasir Assad, Yasir Assad, Ahmed Vandy.

**Formal analysis:** Nasir Assad, Nouman Ahmad, Muhammad Nauman Khan.

**Investigation:** Hansa Gul, Nasir Assad.

**Methodology:** Hansa Gul, Nasir Assad, Ahmed Vandy.

**Validation:** Zahida Nasreen, Muhammad Nauman Khan, Michael Lahai, Sezai Ercişli.

**Visualization:** Nasir Assad, Zahida Nasreen.

**Writing – original draft:** Hansa Gul, Nouman Ahmad, Yasir Assad, Muhammad Nauman Khan.

**Writing – review & editing:** Hansa Gul, Nasir Assad, Zahida Nasreen, Nouman Ahmad, Yasir Assad, Muhammad Nauman Khan, Ahmed Vandy, Michael Lahai, Muhammad Naeem-ul-Hassan, Sezai Ercişli.

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
