## [Decision Letter · Decision Letter 0]

22 Oct 2025

Dear Dr. Vandy,

Thank you for submitting your manuscript to PLOS ONE. After careful consideration, we feel that it has merit but does not fully meet PLOS ONE’s publication criteria as it currently stands. Therefore, we invite you to submit a revised version of the manuscript that addresses the points raised during the review process.

The submission reflects scientific relevance. However, some fundamental issues limit its quality for publication in the current form. For instance, the authors need to justify the significance of the study, relate it to the literature and identify the gap in the existing knowledge that this study aims to address, and ensure an adequate validation of the theoretical analyses reported. Again, what are the limitations of this study, and how can the authors recommend future research on the study? Moreover, some serious concerns have been raised by the reviewers affecting pivotal sections of the study. Kindly pay close attention to these and address them critically before resubmission.

We look forward to receiving your revised manuscript.

Kind regards,

Yusuf Oloruntoyin Ayipo, Ph.D

Academic Editor

PLOS ONE

**Journal Requirements:**

1. When submitting your revision, we need you to address these additional requirements. Please ensure that your manuscript meets PLOS ONE's style requirements, including those for file naming. The PLOS ONE style templates can be found at https://journals.plos.org/plosone/s/file?id=wjVg/PLOSOne_formatting_sample_main_body.pdf and https://journals.plos.org/plosone/s/file?id=ba62/PLOSOne_formatting_sample_title_authors_affiliations.pdf 2. Please include a complete copy of PLOS’ questionnaire on inclusivity in global research in your revised manuscript. Our policy for research in this area aims to improve transparency in the reporting of research performed outside of researchers’ own country or community. The policy applies to researchers who have travelled to a different country to conduct research, research with Indigenous populations or their lands, and research on cultural artefacts. The questionnaire can also be requested at the journal’s discretion for any other submissions, even if these conditions are not met.  Please find more information on the policy and a link to download a blank copy of the questionnaire here: https://journals.plos.org/plosone/s/best-practices-in-research-reporting. Please upload a completed version of your questionnaire as Supporting Information when you resubmit your manuscript. 3. In your Methods section, please provide additional information regarding the permits you obtained for the work. Please ensure you have included the full name of the authority that approved the field site access and, if no permits were required, a brief statement explaining why. 4. Please note that PLOS One has specific guidelines on code sharing for submissions in which author-generated code underpins the findings in the manuscript. In these cases, we expect all author-generated code to be made available without restrictions upon publication of the work. Please review our guidelines at https://journals.plos.org/plosone/s/materials-and-software-sharing#loc-sharing-code and ensure that your code is shared in a way that follows best practice and facilitates reproducibility and reuse. 5. Please upload a new copy of Figures 6, 8, 10 and 11, as the detail is not clear. Please follow the link for more information:  https://journals.plos.org/plosone/s/figures 6. Please include captions for your Supporting Information files at the end of your manuscript, and update any in-text citations to match accordingly. Please see our Supporting Information guidelines for more information: http://journals.plos.org/plosone/s/supporting-information. 7. If the reviewer comments include a recommendation to cite specific previously published works, please review and evaluate these publications to determine whether they are relevant and should be cited. There is no requirement to cite these works unless the editor has indicated otherwise. 

**Additional Editor Comments:**

The submission reflects scientific relevance. However, some fundamental issues limit its quality for publication in the current form. For instance, the authors need to justify the significance of the study, relate it to the literature and identify the gap in the existing knowledge that this study aims to address, and ensure an adequate validation of the theoretical analyses reported. Again, what are the limitations of this study, and how can the authors recommend future research on the study? Moreover, some serious concerns have been raised by the reviewers affecting pivotal sections of the study. Kindly pay close attention to these and address them critically before resubmission.

Reviewers' comments:

**Comments to the Author**

1. Is the manuscript technically sound, and do the data support the conclusions?

Reviewer #1: Yes

Reviewer #2: Yes

Reviewer #3: Partly

Reviewer #4: Yes

Reviewer #5: Yes

2. Has the statistical analysis been performed appropriately and rigorously?

Reviewer #1: No

Reviewer #2: N/A

Reviewer #3: Yes

Reviewer #4: Yes

Reviewer #5: No

3. Have the authors made all data underlying the findings in their manuscript fully available?

Reviewer #1: Yes

Reviewer #2: No

Reviewer #3: Yes

Reviewer #4: Yes

Reviewer #5: No

4. Is the manuscript presented in an intelligible fashion and written in standard English?

Reviewer #1: Yes

Reviewer #2: Yes

Reviewer #3: Yes

Reviewer #4: Yes

Reviewer #5: Yes

**Reviewer #1:**  The manuscript provides a comprehensive and original study of Withania coagulans, integrating phytochemical profiling with computational validation. Although the computational study is well described, the microbiological analysis lacks information. Please, see my comments below:

1. Introduction

Consider highlighting in the introduction how this study advances beyond previous work on W. coagulans and terpenoids, to further emphasize its novelty.

2. Materials and Methods

a) Ensure that all assay conditions (e.g., concentrations, incubation times, controls) are explicitly stated for each experiment.

b) Mention which guidelines (CLSI or EUCAST) are followed to interpret the zone of inhibition. Consider carrying out MIC and MBC assay.

c) The number of replicates for each sample and control (e.g., biological replicates, technical replicates) is not mentioned. This should be clearly stated to ensure statistical relevance.

d) The study does not address the potential cytotoxicity of compounds, which is vital for clinical applications.

e) Consider performing in-vivo antidiabetic assay on mice model, and if possible, carryout histopathological test.

3. Results

Compare the results of antibacterial assay with standard in the light of CLSI and/or EUCAST guidelines

4. Discussions

a) The discussion could be expanded to further explore the broader implications of the findings, including potential applications in drug development and how these compounds compare to existing therapeutics.

b) Consider discussing any limitations of the study and possible directions for future research.

Suggestions:

•Revise sections with lengthy or complex sentences to improve clarity and readability.

•Ensure consistency in terminology throughout the manuscript.

•Expand figure and table legends to be fully self-explanatory.

**Reviewer #2:** The integration of phytochemical profiling, biological assays, and computational modeling addresses a relevant topic in medicinal-plant research. The authors attempt to connect in-vitro results with in-silico validation. The selection of Withania coagulans, a well-known ethnomedicinal species, is of genuine pharmacological interest and aligns with the journal’s interdisciplinary scope.

Following are my specific comments for the authors:

The extraction procedure is well described, but the authors should report the extraction yield (% w/w), specify the working extract concentration, and briefly justify the choice of water as the solvent. This information is helpful for reproducibility and comparison with earlier studies.

The inclusion of at least one Gram-positive strain is suggested (not required though). But the inhibition zones data should be presented with means ± SD from triplicates and statistically compared to the control.

The starch hydrolysis assay based on inhibition zone measurement is unconventional. Please add brief rationale and citation.

The ADMET table gives nearly identical results for all compounds despite their structural diversity? A comment.

The paper repeatedly describes the two terpenoids as “multi-target inhibitors” and “potential drug candidates.” One would be more cautious as the study is limited to preliminary in-vitro assays and in-silico predictions.

The discussion also includes statements, for example, assigning cardiovascular or anticancer properties to compounds that were neither tested nor cited from primary literature. Wherever possible, the authors should avoid extrapolations beyond the evidence.

Units (µg/mL) and decimal formats should be standardized throughout.

In Section 2.7.5, the authors describe DFT optimization at the B3LYP level but do not report the basis set used or confirm that optimized geometries were minima by verifying the absence of imaginary frequencies. It is essential to ensure that the optimized structures correspond to true ground-state minima on the potential energy surface. The authors are suggested to report the number of imaginary frequencies (NImag = 0) for each structure and provide xyz coordinates or fchk files of the optimized geometries.

It is suggested that the below mentioned foundational sources would enrich and depth author's discussion on hardness, softness, chemical potential and reactivity based on DFT.

-Muya, Jules Tshishimbi, et al. "Conceptual DFT study of the chemical reactivity of four natural products with anti-sickling activity." SN Applied Sciences 1.11 (2019): 1457.

-Akintemi, Eric O., Krishna K. Govender, and Thishana Singh. "A DFT study of the chemical reactivity properties, spectroscopy and bioactivity scores of bioactive flavonols." Computational and Theoretical Chemistry 1210 (2022): 113658.

-Kour, Manjinder, Raakhi Gupta, and Raj K. Bansal. "Experimental and theoretical investigation of the reaction of secondary amines with maleic anhydride." Australian Journal of Chemistry 70.12 (2017): 1247-1253.

-Hussein, Y. T., & Azeez, Y. H. (2023). DFT analysis and in silico exploration of drug-likeness, toxicity prediction, bioactivity score, and chemical reactivity properties of the urolithins. Journal of Biomolecular Structure and Dynamics, 41(4), 1168-1177.

Figures need higher resolution and consistent labeling of axes and units. Figures quality at present is very poor and not up to the standards.

Once these issues are addressed, the work could make a useful contribution to the literature on Withania coagulans phytochemistry and bioactivity.

**Reviewer #3:** General assessment:

This manuscript investigates the bioactive potential of an aqueous stem extract of Withania coagulans through GC–MS analysis, antibacterial and antidiabetic assays, antioxidant tests, and extensive computational modeling (docking, ADMET, DFT, MD, and MM/GBSA). The study is well-intentioned and methodologically diverse, addressing a relevant topic in phytochemistry and natural product drug discovery. However, several claims exceed the support provided by the data, and methodological details, especially regarding computational work and data availability, require major clarification. The paper shows considerable potential but requires revision before it can be considered for publication.

Major Comments

1. Experimental confirmation of lead compounds

o The claim that caryophyllene oxide and 2,2-dimethyl-3-(…)-oxirane are the active “multi-target inhibitors” is not fully supported by the data.

o GC–MS identification in a crude extract does not demonstrate biological causality. The authors should either:

Isolate and bioassay the pure compounds, or

Purchase authentic standards and determine their MIC/MBC, IC₅₀, and EC₅₀ values in the same antibacterial, α-amylase, and antioxidant assays.

o Without these tests, conclusions should be limited to in silico predictions of possible leads.

2. Antibacterial and enzyme-inhibition data

o The agar-well diffusion assay alone is insufficient to establish antibacterial potency. Please provide MIC and MBC values using broth microdilution.

o For α-amylase inhibition, report dose–response curves and IC₅₀ values with clear replicate numbers, error bars, and statistical tests.

3. GC–MS identification confidence

o Specify library match scores, retention-index confirmation, and whether derivatization was applied.

o Discuss the limitations of GC–MS for aqueous extracts and, if possible, confirm key compounds with HRMS or co-injection with standards.

4. Computational methods and reproducibility

o Several MD parameters appear inconsistent (e.g., very short equilibration, 0.00 M NaCl, unclear cut-offs). Provide full GROMACS parameter (.mdp) files, grid coordinates, and detailed docking parameters.

o Report how docking poses were validated (e.g., RMSD of redocked co-crystal ligands).

5. Interpretation and tone of claims

o Revise language to avoid causal overstatement (e.g., replace “are multi-target inhibitors” with “are predicted multi-target inhibitors present in the extract”).

o Differentiate clearly between experimental evidence and computational prediction.

Minor Comments

1. Clarify concentrations and volumes used in agar diffusion wells and enzyme assays.

2. Report biological and technical replicate numbers for all assays and include the statistical methods used.

3. Expand figure legends to indicate sample size and significance levels.

4. Correct typographical errors and unit inconsistencies in the MD and results sections.

5. Ensure all software tools are cited with versions and references.

6. Include detailed step-by-step protocols for extraction, assays, and computational analyses as supplementary files.

Recommendation

Decision: Major Revision (encouraged to resubmit)

The manuscript has merit and could make a meaningful contribution once the above issues, especially experimental validation of lead compounds, reproducibility of computational methods, and data deposition, are fully addressed.

**Reviewer #4:** The manuscript would benefit from several revisions to improve consistency, methodological clarity, and alignment with journal standards. Please standardize equation numbering and placement so that references in the text (e.g., “(Eq. 1)”) correspond unambiguously to right-aligned, sequentially labeled equations. In the Introduction, adopt a single citation style; for example, use bracketed, comma-separated numerals throughout rather than mixing semicolons and commas. Define deionized water at first mention as “deionized (DI) water” and use “DI water” consistently instead of “DW.”

In the GC–MS section, the current description of aqueous extract analysis does not address volatility of polar constituents; either include a nonpolar fractionation step prior to GC–MS or add a derivatization protocol (reagent, temperature, time, and any cleanup) suitable for the analytes reported. For the antibacterial assay, erythromycin at 30 µg is a weak comparator for Gram-negative organisms; replace or supplement it with a CLSI-concordant control for Gram-negatives (e.g., ciprofloxacin 5 µg disk, gentamicin 10 µg, or imipenem) and interpret results per CLSI M100. Verify the Salmonella ATCC designation and serovar (e.g., commonly S. Typhimurium ATCC 14028) and update all text and figure labels accordingly. Report diffusion assay units in the appropriate format—µg per disk for disk diffusion, or mass per well (µg) together with the loaded volume (µL) for well diffusion—and include inoculum standardization, agar depth, incubation conditions, and vehicle controls.

For the α-amylase assays, clarify the DNSA control definitions and the inhibition equation explicitly (A_control = enzyme + substrate without inhibitor; A_sample = enzyme + substrate + test item; % inhibition = [(A_control − A_sample)/A_control] × 100), and reconcile these definitions with what is plotted in the figures. The starch diffusion assay incubation of 72 h at 37 °C is atypically long for an enzyme plate and risks denaturation or contamination; shorten to a validated, substantially shorter period and justifying the chosen time. In the protein-preparation workflow, specify whether catalytically or structurally relevant crystallographic waters and cofactors within a defined distance of the binding site were retained, describe protonation state assignment at pH 7.0–7.4, and indicate how histidine tautomers were resolved.

Finally, moderate structure–activity assertions to match analytical confidence. Replace unsupported statements assigning anticancer activity to compounds such as propylamine or methyl ethyl cyclopentene with conservative language (e.g., that certain terpenoids/phenolics have reported cytotoxic activities in the literature and that the present identifications are tentative and require validation). Remove attributions involving Dimefox from any therapeutic discussion or explicitly exclude such tentative matches from interpretation pending confirmation with authentic standards and retention indices. Implementing these revisions will improve reproducibility, interpretability, and compliance with field conventions.

**Reviewer #5:**  This manuscript titled “Identification of Two Terpenoids from Withania coagulans as Multi-Target Inhibitors: An In Vitro and In Silico Study” explores the pharmacological potential of Withania coagulans through a combination of experimental and computational methods. The topic is scientifically relevant and aligns with current interests in plant-based therapeutics and integrated computational validation. However, the manuscript requires substantial improvements in methodological detail, analytical rigor, and presentation to meet publication standards. The following comments address key technical and structural issues aimed at enhancing its clarity, reproducibility, and scientific quality.

Provide quantitative statistics for all biological assays (mean ± SD, ANOVA, or t-test results).

Include docking validation (redocking RMSD ≤ 2.0 Å) and specify detailed grid parameters.

Expand ADMET results with full descriptors and cross-validate using at least one additional predictive platform.

Report MMGBSA energies numerically (ΔG_total, Evdw, Eele, Gsolv) and include confidence intervals.

Clarify the DFT interpretation by distinguishing between stability (large energy gap) and reactivity (small energy gap).

Enhance reproducibility by including molecular simulation input parameters and convergence criteria.

Correlate computational outputs with biochemical validation (e.g., enzyme inhibition constants) to strengthen the mechanistic foundation.

**Do you want your identity to be public for this peer review?** For information about this choice, including consent withdrawal, please see our Privacy Policy

Reviewer #1: No

Reviewer #2: No

Reviewer #3: No

Reviewer #4: No

Reviewer #5: No

---

## [Author Response · Author response to Decision Letter 1]

14 Dec 2025

Dear Reviewers and Editorial Team,

We sincerely thank the Academic Editor and all the reviewers for their time, valuable feedback, and insightful comments on our manuscript. We appreciate the opportunity to revise our work and address the critical points that were raised.

The review process has significantly helped us improve the scientific quality, clarity, and presentation of our study. We have carefully revised the manuscript and responded to each comment point-by-point. We are grateful for your guidance and constructive suggestions, which contributed to refining the methodology, strengthening the interpretation of results, and improving overall transparency and reproducibility.

We hope the revised version meets the journal’s standards and expectations, and we look forward to the possibility of publication in PLOS ONE.

PONE-D-25-53146

Identification of Two Terpenoids from Withania coagulans as Multi-Target Inhibitors: An In Vitro and In Silico Study

PLOS ONE

Dear Dr. Vandy,

Thank you for submitting your manuscript to PLOS ONE. After careful consideration, we feel that it has merit but does not fully meet PLOS ONE’s publication criteria as it currently stands. Therefore, we invite you to submit a revised version of the manuscript that addresses the points raised during the review process.

The submission reflects scientific relevance. However, some fundamental issues limit its quality for publication in the current form. For instance, the authors need to justify the significance of the study, relate it to the literature and identify the gap in the existing knowledge that this study aims to address, and ensure an adequate validation of the theoretical analyses reported. Again, what are the limitations of this study, and how can the authors recommend future research on the study? Moreover, some serious concerns have been raised by the reviewers affecting pivotal sections of the study. Kindly pay close attention to these and address them critically before resubmission.

We look forward to receiving your revised manuscript.

Kind regards,

Yusuf Oloruntoyin Ayipo, Ph.D

Academic Editor

PLOS ONE

Journal Requirements:

Authors response: Thank you for the comment. We have reshaped the manuscript formatting to meets PLOS ONE's style requirements. Font size and Figure labeling has been revised.

Authors response: The form has been completed and uploaded together with the submission.

Authors response: This study did not involve human participants, indigenous communities, or the collection of materials requiring access or benefit-sharing agreements. Plant stems of Withania coagulans (Dunal) were collected from the Cholistan Desert, Punjab, Pakistan, on privately owned land belonging to the authors, where the species grows naturally. Botanical authentication was conducted by Muhammad Numan Khan, Department of Botany, Islamia College Peshawar. As the collection occurred on private property and did not involve protected or endangered species, no permits or licenses were required.

4. Please note that PLOS One has specific guidelines on code sharing for submissions in which author-generated code underpins the findings in the manuscript. In these cases, we expect all author-generated code to be made available without restrictions upon publication of the work. Please review our guidelines at https://journals.plos.org/plosone/s/materials-and-software-sharing#loc-sharing-code and ensure that your code is shared in a way that follows best practice and facilitates reproducibility and reuse.

Authors response: The raw data has been uploaded with the submission.

5. Please upload a new copy of Figures 6, 8, 10 and 11, as the detail is not clear. Please follow the link for more information: https://journals.plos.org/plosone/s/figures

Authors response: Done as requested, Thank you.

Authors response: Done as requested, Thank you.

Authors response: Thank you for the instruction

Additional Editor Comments:

The submission reflects scientific relevance. However, some fundamental issues limit its quality for publication in the current form. For instance, the authors need to justify the significance of the study, relate it to the literature and identify the gap in the existing knowledge that this study aims to address, and ensure an adequate validation of the theoretical analyses reported. Again, what are the limitations of this study, and how can the authors recommend future research on the study? Moreover, some serious concerns have been raised by the reviewers affecting pivotal sections of the study. Kindly pay close attention to these and address them critically before resubmission.

Authors response: Thank you for the comments, the manuscript has now been revised.

Authors response:

Reviewers' comments:

Reviewer's Responses to Questions

Comments to the Author

1. Is the manuscript technically sound, and do the data support the conclusions?

Reviewer #1: Yes

Reviewer #2: Yes

Reviewer #3: Partly

Reviewer #4: Yes

Reviewer #5: Yes

2. Has the statistical analysis been performed appropriately and rigorously?

Reviewer #1: No

Reviewer #2: N/A

Reviewer #3: Yes

Reviewer #4: Yes

Reviewer #5: No

3. Have the authors made all data underlying the findings in their manuscript fully available?

Reviewer #1: Yes

Reviewer #2: No

Reviewer #3: Yes

Reviewer #4: Yes

Reviewer #5: No

4. Is the manuscript presented in an intelligible fashion and written in standard English?

Reviewer #1: Yes

Reviewer #2: Yes

Reviewer #3: Yes

Reviewer #4: Yes

Reviewer #5: Yes

5. Review Comments to the Author

Reviewer #1

Reviewer #1: The manuscript provides a comprehensive and original study of Withania coagulans, integrating phytochemical profiling with computational validation. Although the computational study is well described, the microbiological analysis lacks information. Please, see my comments below:

1. Introduction

Consider highlighting in the introduction how this study advances beyond previous work on W. coagulans and terpenoids, to further emphasize its novelty.

Authors Response: We thank the reviewer for this insightful suggestion. In response, we have revised the Introduction to more clearly highlight how the present study advances beyond previous work on Withania coagulans and its terpenoid constituents. The revised text emphasizes the existing gaps in the literature particularly the limited characterization of terpenoids and the absence of integrated in vitro–in silico approaches and explains how our study fills these gaps.

2. Materials and Methods

a) Ensure that all assay conditions (e.g., concentrations, incubation times, controls) are explicitly stated for each experiment.

Authors response: We thank the reviewer for this important comment. We have carefully revised the Materials and Methods section to ensure that all assay conditions—including concentrations, volumes, incubation times, temperatures, and control details—are clearly and explicitly described for each experiment.

b) Mention which guidelines (CLSI or EUCAST) are followed to interpret the zone of inhibition. Consider carrying out MIC and MBC assay.

Authors response: Thank you for this insightful suggestion. We have updated (Antibacterial Assay) to clearly state that zone of inhibition results were interpreted according to the CLSI M100 (2020) guidelines. Additionally, we have now performed MIC and MBC assays using the broth microdilution method following CLSI M07-A9 protocols, as recommended.

c) The number of replicates for each sample and control (e.g., biological replicates, technical replicates) is not mentioned. This should be clearly stated to ensure statistical relevance.

Authors response: We thank the reviewer for this important observation. We have now clarified the number of replicates used in all experiments.

d) The study does not address the potential cytotoxicity of compounds, which is vital for clinical applications.

Authors response: We thank the reviewer for this important observation. We acknowledge that assessing cytotoxicity is critical for evaluating the clinical relevance of the identified terpenoids. While the current study focused on phytochemical profiling, in vitro antibacterial, antidiabetic, and antioxidant activity, and in silico analyses, we did not perform experimental cytotoxicity assays.

We have now included a statement in the Discussion and Limitations sections acknowledging this limitation and indicating that future studies will incorporate cytotoxicity testing using appropriate mammalian cell lines to evaluate safety and therapeutic potential. This ensures transparency and aligns with clinical relevance considerations.

e) Consider performing in-vivo antidiabetic assay on mice model, and if possible, carryout histopathological test.

Authors response: We sincerely thank the reviewer for this valuable suggestion. We fully acknowledge that in vivo antidiabetic evaluation and histopathological analysis would provide important insights into the therapeutic potential and safety of the identified compounds.

However, due to facility limitations and the lack of ethical approval for animal experiments at the current stage, we were unable to perform these assays. We have noted this as a limitation in the manuscript and plan to include in vivo antidiabetic studies and histopathological analysis in future work as part of our ongoing research. We appreciate the reviewer’s understanding and constructive feedback.

3. Results

Compare the results of antibacterial assay with standard in the light of CLSI and/or EUCAST guidelines

Authors response: We thank the reviewer for this valuable suggestion. The antibacterial assay results have now been interpreted according to the CLSI (M100) guidelines. Compared with the standard antibiotic (erythromycin), Withania coagulans extract exhibited a concentration-dependent inhibitory effect against Salmonella Typhi and Escherichia coli.

4. Discussions

a) The discussion could be expanded to further explore the broader implications of the findings, including potential applications in drug development and how these compounds compare to existing therapeutics.

Authors response: We thank the reviewer for this constructive suggestion. The Discussion section has been expanded to better highlight the broader implications of our findings in the context of drug development.

b) Consider discussing any limitations of the study and possible directions for future research.

Suggestions:

Authors response: We thank the reviewer for highlighting this point. We have now included a clear discussion of the study’s limitations and outlined directions for future research

•Revise sections with lengthy or complex sentences to improve clarity a

---

## [Decision Letter · Decision Letter 1]

22 Jan 2026

Dear Dr. Vandy,

Thank you for submitting your manuscript to PLOS ONE. After careful consideration, we feel that it has merit but does not fully meet PLOS ONE’s publication criteria as it currently stands. Therefore, we invite you to submit a revised version of the manuscript that addresses the points raised during the review process.

**ACADEMIC EDITOR:**

We look forward to receiving your revised manuscript.

Kind regards,

Yusuf Oloruntoyin Ayipo, Ph.D

Academic Editor

PLOS One

**Journal Requirements:**

**Additional Editor Comments:**

Understandably, the authors have responded positively to the previous concerns. The revision has improved the quality of the submission significantly. However, some major points still deserve a substantial attention of the authors as specifically pointed out by Reviewer #1. I hereby recommend another round of revision to address these.

Reviewers' comments:

Reviewer's Responses to Questions

**Comments to the Author**

Reviewer #1: All comments have been addressed

Reviewer #2: All comments have been addressed

Reviewer #3: All comments have been addressed

Reviewer #4: All comments have been addressed

2. Is the manuscript technically sound, and do the data support the conclusions?

Reviewer #1: Yes

Reviewer #2: Yes

Reviewer #3: Yes

Reviewer #4: Yes

3. Has the statistical analysis been performed appropriately and rigorously?

Reviewer #1: No

Reviewer #2: Yes

Reviewer #3: Yes

Reviewer #4: Yes

4. Have the authors made all data underlying the findings in their manuscript fully available?

Reviewer #1: No

Reviewer #2: Yes

Reviewer #3: Yes

Reviewer #4: Yes

5. Is the manuscript presented in an intelligible fashion and written in standard English?

Reviewer #1: Yes

Reviewer #2: Yes

Reviewer #3: Yes

Reviewer #4: Yes

Reviewer #1: The authors present a multidisciplinary study integrating phytochemical profiling (GC–MS), in vitro biological assays (antibacterial, antidiabetic, antioxidant), and extensive computational validation (docking, ADMET, DFT, MD simulations, MM/GBSA) to investigate the pharmacological potential of Withania coagulans stem extract, with a focus on two terpenoids. The revised submission addresses most of the major and minor concerns raised in the previous round of review, resulting in a significant improvement in both scientific rigor and presentation.

Provided the following edits and clarifications are made, it will be suitable for publication-

1) While the integration is new for W. coagulans, similar strategies have been applied to other medicinal plants, making the contribution incremental rather than transformative. In the introduction, explicitly highlight how this work advances beyond previous studies on W. coagulans and terpenoids.

2) The manuscript could better articulate how its findings compare to existing therapeutics and the broader implications for drug development. Expand the discussion to compare findings with current antimicrobial agents and discuss the potential for clinical translation.

3) The methods section lacks critical detail, especially for the antimicrobial assays, making it difficult to reproduce the work.

4) Table 3 omits standard deviations for zone of inhibition measurements, despite the legend stating data are presented as mean ± SD. Apply appropriate statistical tests, and present results as mean ± standard deviation with corresponding p-values.

5) The manuscript does not describe the use of standard antibiotics (e.g., erythromycin) in the MIC and MBC assays. Include MIC and MBC values for these standards in Table 4 to enable direct comparison with test compounds. Apply appropriate statistical tests and report all data as mean ± standard deviation with p-values.

Reviewer #2: While all the reviewers comments has been addressed, there are still some formatting improvements pending mainly consistency. For example, ΔE_gap versus ΔE_Gap, ΔG_TOTAL versus ΔGTOTAL and so on. Please make sure to make these things carefully consistent while proofreading.

Reviewer #3: The article strength lies in its transparent and methodologically rigorous integration of phytochemical profiling, in vitro biological screening, and validated in silico analyses, including molecular docking, molecular dynamics simulations, MM/GBSA binding energetics, and DFT calculations. The manuscript demonstrates strong reproducibility through detailed protocol reporting, validated computational workflows, and full data availability in compliance with PLOS ONE standards. By focusing on underexplored terpenoid constituents of Withania coagulans, the work provides a sound, hypothesis-generating contribution to medicinal-plant research while maintaining appropriate interpretive caution.

Reviewer #4: The English used is very much correct. but i believe the author should cross check all to identify where there is a typographiccal error and correct it all.

**Do you want your identity to be public for this peer review?** For information about this choice, including consent withdrawal, please see our Privacy Policy

Reviewer #1: **Yes:** Muhammad Salehuddin Ayubee

Reviewer #2: No

Reviewer #3: No

Reviewer #4: **Yes:** Abdullahi Ashimi

---

## [Author Response · Author response to Decision Letter 2]

29 Jan 2026

Dear Editor and the Reviewers,

We would like to express our sincere gratitude for your constructive feedback and for the time you dedicated to reviewing our manuscript titled " Identification of two terpenoids from Withania coagulans with predicted multitarget binding affinity: An in vitro and in silico study". We deeply appreciate your thoughtful and insightful comments, which have significantly improved the quality of our submission.

Additional Editor Comments:

Understandably, the authors have responded positively to the previous concerns. The revision has improved the quality of the submission significantly. However, some major points still deserve a substantial attention of the authors as specifically pointed out by Reviewer #1. I hereby recommend another round of revision to address these.

Authors response: Thank you for your constructive feedback and for recognizing the improvements made in the revised manuscript. We sincerely appreciate the time and effort you and Reviewer #1 have dedicated to reviewing our work.

Unarguably, we agree to the points highlighted by Reviewer #1. We want to assure you that we have carefully reviewed these concerns and have made the necessary revisions to address them in detail. All feedback has been thoroughly incorporated into the manuscript, and we believe the updated version now meets the expectations of the reviewers.

We appreciate your recommendation for another round of revision and are confident that the changes made will significantly improve the quality of the submission.

Thank you once again for your guidance. We look forward to your further feedback.

Reviewers' comments:

Reviewer #1:

The authors present a multidisciplinary study integrating phytochemical profiling (GC–MS), in vitro biological assays (antibacterial, antidiabetic, antioxidant), and extensive computational validation (docking, ADMET, DFT, MD simulations, MM/GBSA) to investigate the pharmacological potential of Withania coagulans stem extract, with a focus on two terpenoids. The revised submission addresses most of the major and minor concerns raised in the previous round of review, resulting in a significant improvement in both scientific rigor and presentation.

Provided the following edits and clarifications are made, it will be suitable for publication-

1) While the integration is new for W. coagulans, similar strategies have been applied to other medicinal plants, making the contribution incremental rather than transformative. In the introduction, explicitly highlight how this work advances beyond previous studies on W. coagulans and terpenoids.

Authors response; Thank you for your constructive feedback. In response to your comment regarding the novelty of the study, we have revised the introduction to explicitly highlight how this work advances beyond previous studies on Withania coagulans. In this study, for the first time, we explore terpenoid compounds in W. coagulans beyond the well-known withanolides, revealing a new dimension of the plant’s medicinal potential. This approach uncovers previously unreported terpenoids and their multi-functional properties, particularly their promising applications in anticancer strategies. These points have been added and revised in the introduction to emphasize the novelty and significance of our findings.

2) The manuscript could better articulate how its findings compare to existing therapeutics and the broader implications for drug development. Expand the discussion to compare findings with current antimicrobial agents and discuss the potential for clinical translation.

Authors response; Thank you for your valuable feedback. In response to your suggestion, we have expanded the Discussion section to better articulate how our findings compare to existing antimicrobial agents and their broader implications for drug development.

3) The methods section lacks critical detail, especially for the antimicrobial assays, making it difficult to reproduce the work.

Authors response; Thank you for your valuable feedback regarding the methods section. In response, we have revised and updated the methodology to provide more detailed and reproducible instructions, particularly for the antibacterial activity assay.

4) Table 3 omits standard deviations for zone of inhibition measurements, despite the legend stating data are presented as mean ± SD. Apply appropriate statistical tests, and present results as mean ± standard deviation with corresponding p-values.

Authors response; Thank you for your constructive feedback regarding Table 3 and the inclusion of standard deviations (SD) and statistical analysis. In response to your comment, we have updated Table 3 to include the standard deviations (SD) for the zone of inhibition measurements and the corresponding p-values for statistical significance.

5) The manuscript does not describe the use of standard antibiotics (e.g., erythromycin) in the MIC and MBC assays. Include MIC and MBC values for these standards in Table 4 to enable direct comparison with test compounds. Apply appropriate statistical tests and report all data as mean ± standard deviation with p-values.

Authors response; Thank you for your valuable feedback. In response to your comment, we have now included the MIC and MBC values for erythromycin (a standard antibiotic) in Table 4, allowing for a direct comparison with the test compounds, such as Withania coagulans extract.

Reviewer #2:

While all the reviewers comments has been addressed, there are still some formatting improvements pending mainly consistency. For example, ΔE_gap versus ΔE_Gap, ΔG_TOTAL versus ΔGTOTAL and so on. Please make sure to make these things carefully consistent while proofreading.

Authors response; Thank you dear reviewer: Thoroughly revised and updated.

Reviewer #3:

The article strength lies in its transparent and methodologically rigorous integration of phytochemical profiling, in vitro biological screening, and validated in silico analyses, including molecular docking, molecular dynamics simulations, MM/GBSA binding energetics, and DFT calculations. The manuscript demonstrates strong reproducibility through detailed protocol reporting, validated computational workflows, and full data availability in compliance with PLOS ONE standards. By focusing on underexplored terpenoid constituents of Withania coagulans, the work provides a sound, hypothesis-generating contribution to medicinal-plant research while maintaining appropriate interpretive caution.

Authors response; Thank you for your positive feedback. We appreciate your recognition of the rigorous integration of phytochemical profiling, in vitro screening, and in silico analyses in our study. We are also glad that you found our focus on underexplored terpenoids in Withania coagulans to be a valuable contribution to medicinal-plant research. Your encouraging comments are greatly appreciated.

Reviewer #4:

The English used is very much correct. but i believe the author should cross check all to identify where there is a typographiccal error and correct it all.

Authors response; Thank you dear reviewer: Thoroughly revised and updated.

Comments to editor: We have carefully addressed all the reviewer comments and made the necessary revisions to the manuscript. The changes are clearly marked in red text, and blue text highlights the grammatical corrections. We believe these revisions have strengthened the manuscript, and we are confident that it now meets the publication standards. We appreciate your continued guidance and look forward to your further feedback.

---

## [Editor Report · Decision Letter 2]

3 Feb 2026

Identification of two terpenoids from Withania coagulanswith predicted multitarget binding affinity: An in vitro and in silico study

PONE-D-25-53146R2

Dear Dr. Vandy,

We’re pleased to inform you that your manuscript has been judged scientifically suitable for publication and will be formally accepted for publication once it meets all outstanding technical requirements.

Kind regards,

Yusuf Oloruntoyin Ayipo, Ph.D

Academic Editor

PLOS One

Additional Editor Comments (optional):

Congratulations to the authors. The study is timely and well-designed. Again, the submission meets the level of scientific rigour required for publication in this title, and all the concerns raised by the respective reviewers have been addressed satisfactorily. I hereby recommend the manuscript for publication in the current version.
---

## [Editor Report · Acceptance letter]

PONE-D-25-53146R2

PLOS One

Dear Dr. Vandy,

I'm pleased to inform you that your manuscript has been deemed suitable for publication in PLOS One. Congratulations! Your manuscript is now being handed over to our production team.

Kind regards,

on behalf of

Dr. Yusuf Oloruntoyin Ayipo

Academic Editor

PLOS One